# Molecular mechanism for Rabex-5 GEF activation by Rabaptin-5

**Zhe Zhang[1], Tianlong Zhang[1], Shanshan Wang[1], Zhou Gong[2], Chun Tang[2], Jiangye Chen[1], Jianping Ding[1]\***

[1]State Key Laboratory of Molecular Biology, Institute of Biochemistry and Cell Biology, Shanghai Institutes for Biological Sciences, Chinese Academy of Sciences, Shanghai, China; [2]State Key Laboratory of Magnetic Resonance and Atomic and Molecular Physics, Wuhan Institute of Physics and Mathematics, Chinese Academy of Sciences, Wuhan, China

**Abstract** Rabex-5 and Rabaptin-5 function together to activate Rab5 and further promote early endosomal fusion in endocytosis. The Rabex-5 GEF activity is autoinhibited by the Rabex-5 CC domain (Rabex-5CC) and activated by the Rabaptin-5 C2-1 domain (Rabaptin-5C21) with yet unknown mechanism. We report here the crystal structures of Rabex-5 in complex with the dimeric Rabaptin-5C21 (Rabaptin-5C21$_2$) and in complex with Rabaptin-5C21$_2$ and Rab5, along with biophysical and biochemical analyses. We show that Rabex-5CC assumes an amphipathic α-helix which binds weakly to the substrate-binding site of the GEF domain, leading to weak autoinhibition of the GEF activity. Binding of Rabaptin-5C21 to Rabex-5 displaces Rabex-5CC to yield a largely exposed substrate-binding site, leading to release of the GEF activity. In the ternary complex the substrate-binding site of Rabex-5 is completely exposed to bind and activate Rab5. Our results reveal the molecular mechanism for the regulation of the Rabex-5 GEF activity.

**\*For correspondence:** jpding@sibcb.ac.cn

**Competing interests:** The authors declare that no competing interests exist.

## Introduction

Endocytosis is a major process which eukaryotic cells use to absorb extracellular materials (*Doherty and McMahon, 2009*; *Grant and Donaldson, 2009*; *Huotari and Helenius, 2011*). In this process, small GTPase Rab5 functions as a master regulator of the early endosomal biogenesis (*Stenmark, 2009*; *Mizuno-Yamasaki et al., 2012*; *Zeigerer et al., 2012*). Rab5 is localized to early endosomal membrane via its isoprenylated C-terminus and regulates early endosomal fusion through interactions with an array of effectors including Rabaptin-5 (*Stenmark et al., 1995*), Rabenosyn-5 (*Nielsen et al., 2000*), EEA1 (*Mills et al., 1998*; *Simonsen et al., 1998*), PI3Ks (*Li et al., 1995*; *Christoforidis et al., 1999*), and APPLs (*Miaczynska et al., 2004*). Like other small GTPases, Rab5 exists mainly in two states, the GTP-bound active state and the GDP-bound inactive state, and requires guanosine nucleotide exchange factor (GEF) for activation and GTPase-activating protein (GAP) for inactivation.

Rabex-5 is a specific GEF for Rab5, Rab17, and Rab21 (*Horiuchi et al., 1997*; *Delprato et al., 2004*; *Delprato and Lambright, 2007*; *Mori et al., 2013*). The GEF domain is located in the middle and consists of a helical bundle (HB) domain and a Vps9 domain (*Figure 1—figure supplement 1*). Besides, the N-terminal region comprises two distinct ubiquitin-binding domains, a zinc finger domain and a motif interacting with ubiquitin domain, which can interact with ubiquitinated cargoes or adaptors to recruit Rabex-5 to early endosomal membrane (*Lee et al., 2006*; *Mattera et al., 2006*; *Penengo et al., 2006*; *Mattera and Bonifacino, 2008*) and function as an E3 ligase for Ras ubiquitination to promote Ras endosomal localization (*Xu et al., 2010*; *Yan et al., 2010*). The following membrane binding motif domain and the HB domain together compose an early endosomal targeting domain that can direct Rabex-5 to early endosomal membrane (*Zhu et al., 2007*). The C-terminal region consists of a

**eLife digest** Cells need to allow various molecules to pass through the plasma membrane on their surface. Some molecules have to enter the cell, whereas others have to leave. Cells rely on a process called endocytosis to move large molecules into the cell. This involves part of the membrane engulfing the molecule to form a 'bubble' around it. This bubble, which is called an endosome, then moves the molecule to final destination inside the cell.

A protein called Rab5 controls how a new endosome is produced. However, before this can happen, various other molecules—including two proteins called Rabex-5 and Rabaptin-5—must activate the Rab5 protein. Exactly how these three proteins interact with each other was unknown.

Zhang et al. used X-ray crystallography to examine the structures of the complexes formed when Rabex-5 and Rabaptin-5 bind to each other, both when Rab5 is present, and also when it is absent. Biochemical and biophysical experiments confirmed that the Rabex-5/Rabaptin-5 complex is able to activate Rab5.

Zhang et al. also found that Rabex-5, on its own, folds so that the site that normally binds to Rab5 instead binds to a different part of Rabex-5, thus preventing endocytosis. However, when Rabaptin-5 forms a complex with Rabex-5, the Rab5 binding site is freed up.

The Rabex-5/Rabaptin-5 complex can switch between a V shape and a linear structure. Binding to Rab5 stabilizes the linear form of the complex, which then helps activate Rab5, and subsequently the activated Rab5 can interact with other downstream molecules, triggering endocytosis.

coiled-coil (CC) domain and a proline rich region; the CC domain is involved in autoinhibition of the GEF activity and binding of Rabaptin-5 (*Lippe et al., 2001*; *Mattera et al., 2006*; *Delprato and Lambright, 2007*). Rabaptin-5 is a key effector of Rab5 and plays an important role in both homotypic and heterotypic fusions of early endosomes (*Stenmark et al., 1995*; *Stenmark, 2009*). It is a scaffold protein consisting of primarily four coiled-coil domains, namely C1-1, C1-2, C2-1, and C2-2 domains (*Figure 1—figure supplement 1*). The C2-1 domain is responsible for interaction with and recruitment of Rabex-5 to early endosomal membrane to activate Rab5 (*Lippe et al., 2001*; *Mattera et al., 2006*; *Delprato and Lambright, 2007*). Besides, the N-terminal region can mediate interactions with Rab4 and Rab8 (*Vitale et al., 1998*; *Omori et al., 2008*); the middle region can interact with the GAE and GAT domains of GGAs that function as effectors of the Arf family small GTPases in the tethering and fusion of trans Golgi network (TGN) (*Mattera et al., 2003*; *Miller et al., 2003*; *Zhu et al., 2004a*); and the C-terminal region can interact with the GTP-bound Rab5 that recruits Rabaptin-5 to early endosomal membrane (*Vitale et al., 1998*; *Zhu et al., 2004b*). In addition, Rabex-5 and Rabaptin-5 have been shown to function as neoplastic tumor suppressors and are implicated in human cancers (*Magnusson et al., 2001*; *Wang et al., 2009*; *Christoforides et al., 2012*; *Thomas and Strutt, 2014*), and Rabex-5 has also been shown to determine the neurite localization of its substrate Rab proteins and thus plays an important role in the development of hippocampal neurons (*Mori and Fukuda, 2013*; *Mori et al., 2013*).

Previous structural, biochemical, and biological data have demonstrated that Rabex-5 and Rabaptin-5 function together to activate Rab5 in endocytosis; the GEF activity of Rabex-5 could be autoinhibited by its CC domain and activated by binding of the Rabaptin-5 C2-1 domain (Rabaptin-5C21) to the CC domain (*Lippe et al., 2001*; *Delprato and Lambright, 2007*; *Zhu et al., 2007*, *2010*). However, the underlying molecular mechanism is unclear. We report here the crystal structures of a Rabex-5 variant in complex with the dimeric Rabaptin-5C21 (Rabaptin-5C21$_2$) and in complex with Rabaptin-5C21$_2$ and Rab5. The structural data together with the in vitro functional data reveal the molecular mechanism for the regulation of the Rabex-5 GEF activity.

## Results

### Structure of the Rabex-5Δ-Rabaptin-5C21$_2$ complex

To investigate the molecular mechanism of the regulation of the Rabex-5 GEF activity, we were intent to determine the crystal structures of Rabex-5 containing the GEF and CC domains (residues 132–455, Rabex-5) alone and in complex with the Rabaptin-5 C2-1 domain (residues 552–642, Rabaptin-5C21).

We were able to obtain Rabex-5 and the Rabex-5-Rabaptin-5C21 (R2) complex with high purity, stability, and homogeneity, but unfortunately failed to grow any crystals for either Rabex-5 or the R2 complex. Partial digestion of the R2 complex with trypsin shows that Rabex-5 could be proteolyzed in the linker between the GEF and CC domains (*Figure 1—figure supplement 2*), indicating that the linker is surface exposed with high flexibility which may prevent proper crystal packing. Thus, we constructed a series of Rabex-5 variants containing different forms of linker deletion (Rabex-5Δ). Although none could be crystallized alone, one Rabex-5Δ variant (residues 132–455Δ393–407) led to successful crystallization of the Rabex-5Δ-Rabaptin-5C21 (R2Δ) complex.

The crystal structure of the Rabex-5Δ-Rabaptin-5C21 complex was determined at 3.10 Å resolution (*Table 1*), containing one Rabex-5Δ and two Rabaptin-5C21 or one Rabex-5Δ-Rabaptin-5C21$_2$ complex per asymmetric unit (*Figure 1A* and *Figure 1—figure supplement 3A*). The N- and C-terminal regions of each Rabaptin-5C21 form two α-helices linked together by a short loop to assume a 'V' shaped conformation with an inclination angle of about 40°; and the N- and C-terminal α-helices of the two Rabaptin-5C21 dimerize with each other to form two two-helix bundles. In addition, two symmetry-related complexes further dimerize through the N-terminal α-helices of Rabaptin-5C21$_2$ (*Figure 1B*).

Rabex-5CC (residues 413–452) forms a long amphipathic α-helix (about 60 Å) (*Figure 1A-D*) that is in agreement with the prediction by *Delprato and Lambright (2007)*. It packs in parallel with the C-terminal α-helices of Rabaptin-5C21$_2$ to form a tight three-helix bundle with its nonpolar surface buried in a hydrophobic surface groove of Rabaptin-5C21$_2$. The interactions are dominantly hydrophobic that bury a total of solvent accessible surface area of 2664 Å$^2$. The residues that were suggested to be involved in the autoinhibition of the GEF activity, including Asn413, Leu414, Leu417, Leu420, Arg423, and Ile427 (*Delprato and Lambright, 2007*), are located in the N-terminal half of the nonpolar surface, and several of them (Leu420, Arg423, and Ile427) are involved in the interactions with Rabaptin-5C21$_2$ and buried in the interaction interface (*Figure 1C,D*).

The HB domain (residues 132–229) and Vps9 domain (residues 230–368) of the Rabex-5 GEF domain are well defined except a few surface exposed loops (*Figure 1A*). The C-terminal helix αC and the following linker (residues 369–412 with the deleted residues 393–407) are also disordered, consistent with our trypsin digestion results (*Figure 1—figure supplement 2*). The distance between the visible C-terminal end of the GEF domain and N-terminal end of the CC domain is about 10 Å, which is large enough to accommodate the disordered 29 residues with a loop conformation, suggesting that the positions and conformations of the GEF and CC domains are unlikely constrained by the shortened linker.

The overall structure of the GEF domain is very similar to that in the free form (*Delprato et al., 2004*) and in complex with Rab21 (*Delprato and Lambright, 2007*) (RMSD of ~0.90 Å for 228 Cα atoms) (*Figure 1—figure supplement 3B*). At the substrate-binding site, there is a surface groove composed of largely nonpolar residues, which exhibits good chemical and geometrical complementarities with the nonpolar surface of the amphipathic helix of Rabex-5CC (*Figure 1E*). The GEF domain packs along the three-helix bundle formed by Rabex-5CC and Rabaptin-5C21$_2$ (*Figure 1A*). The interactions involve a small portion of the substrate-binding site, a small portion of the N-terminal region of Rabex-5CC adjacent to the nonpolar surface, and a small portion of the N-terminal region of one Rabaptin-5C21 C-terminal α-helix (*Figure 1F*). The interaction interface buries a total of solvent accessible surface area of 1040 Å$^2$, which is much smaller than that between Rabex-5 and Rab21 (2400 Å$^2$) (*Delprato and Lambright, 2007*). Rab21 uses switch I, switch II, and the interswitch region to interact with the substrate-binding site of Rabex-5 (*Delprato and Lambright, 2007*). Although the binding sites for switch II and a small portion of the interswitch region of Rab21 are occupied by the three-helix bundle, the binding sites for switch I and a large portion of the interswitch region of Rab21 are exposed to the solvent (*Figure 1—figure supplement 3C*). Hence, we consider that the substrate-binding site of Rabex-5 is largely exposed to the solvent and partially accessible by the substrate.

To explore the conservations of the conformations of Rabex-5CC and Rabaptin-5C21 and the interactions between Rabex-5CC and Rabaptin-5C21, we also determined the crystal structures of Rabex-5CC alone at 2.00 Å resolution and in complex with Rabaptin-5C21$_2$ at 2.20 Å resolution (*Table 1*). In the Rabex-5CC structure, Rabex-5CC also forms a long α-helix (about 65 Å) and the four Rabex-5CC in the asymmetric unit form a tight four-helix bundle via the nonpolar surface (*Figure 1—figure supplement 4*). In the structure of the Rabex-5CC-Rabaptin-5C21$_2$ complex, the asymmetric unit contains one complex (*Figure 1—figure supplement 5*). Notably, each Rabaptin-5C21 forms a long α-helix (about 125 Å) and two Rabaptin-5C21 dimerize in parallel to form a twisted linear two-helix

**Table 1.** Summary of diffraction data and structure refinement statistics

| | Rabex-5CC | Rabex-5CC-Rabaptin-5C21$_2$ | Rabex-5Δ-Rabaptin-5C21$_2$ | Rab5-Rabex-5Δ-Rabaptin-5C21$_2$ |
|---|---|---|---|---|
| Diffraction data | | | | |
| Wavelength (Å) | 0.9200 | 1.0000 | 0.9793 | 0.9785 |
| Space group | $P2_1$ | C2 | $P3_121$ | $P4_12_12$ |
| Cell parameters | | | | |
| a (Å) | 46.8 | 90.0 | 87.2 | 174.8 |
| b (Å) | 40.3 | 28.9 | 87.2 | 174.8 |
| c (Å) | 51.6 | 108.0 | 168.9 | 149.0 |
| α (°) | 90.0 | 90.0 | 90.0 | 90.0 |
| β (°) | 95.1 | 102.2 | 90.0 | 90.0 |
| γ (°) | 90.0 | 90.0 | 120.0 | 90.0 |
| Resolution (Å) | 50.0–2.00 | 50.0–2.20 | 50.0–3.10 | 50.0–4.60 |
| | (2.07–2.00)* | (2.28–2.20) | (3.21–3.10) | (4.76–4.60) |
| Observed reflections | 38,445 | 47,482 | 79,255 | 124,340 |
| Unique reflections (I/σ(I) > 0) | 12,748 | 13,816 | 13,730 | 12,699 |
| Average redundancy | 3.0 (3.0) | 3.4 (3.0) | 5.8 (6.0) | 9.8 (9.0) |
| Average I/σ(I) | 23.6 (14.0) | 21.2 (3.4) | 20.1 (2.4) | 18.2 (2.8) |
| Completeness (%) | 96.4 (97.7) | 97.7 (85.8) | 98.1 (100.0) | 97.6 (95.8) |
| $R_{merge}$ (%)† | 5.3 (9.3) | 6.0 (28.0) | 8.2 (64.3) | 11.7 (94.3) |
| Refinement and structure model | | | | |
| Reflections (Fo ≥ 0σ(Fo)) | | | | |
| Working set | 11,437 | 12,433 | 10,806 | 11,982 |
| Test set | 622 | 691 | 601 | 631 |
| $R_{work}/R_{free}$ (%)‡ | 19.1/23.4 | 19.3/23.5 | 26.4/31.5 | 25.1/34.3 |
| No. of atoms | 1726 | 1814 | 3074 | 9679 |
| Protein | 1621 | 1627 | 3074 | 9679 |
| Water | 105 | 187 | – | – |
| Average B factor (Å$^2$) | | | | |
| All atoms | 22.8 | 58.5 | 72.0 | 187.3 |
| Main-chain atoms | 17.9 | 51.8 | 72.6 | 186.9 |
| Side-chain atoms | 25.9 | 64.0 | 70.7 | 187.7 |
| Water | 34.9 | 63.5 | - | - |
| RMS deviations | | | | |
| Bond lengths (Å) | 0.018 | 0.014 | 0.005 | 0.015 |
| Bond angles (°) | 1.61 | 1.37 | 1.27 | 1.87 |
| Ramachandran plot (%) | | | | |
| Most favored | 99.5 | 99.5 | 92.1 | 93.8 |
| Allowed | 0.5 | 0.5 | 7.6 | 5.8 |
| Generously allowed | 0.0 | 0.0 | 0.3 | 0.5 |

*Numbers in parentheses represent the highest resolution shell.

†Rmerge = ∑hkl∑i|Ii(hkl)i−<I(hkl)>|/∑hkl∑iIi(hkl).

‡R = ∑hkl||Fo|−|Fc||/∑hkl|Fo|.

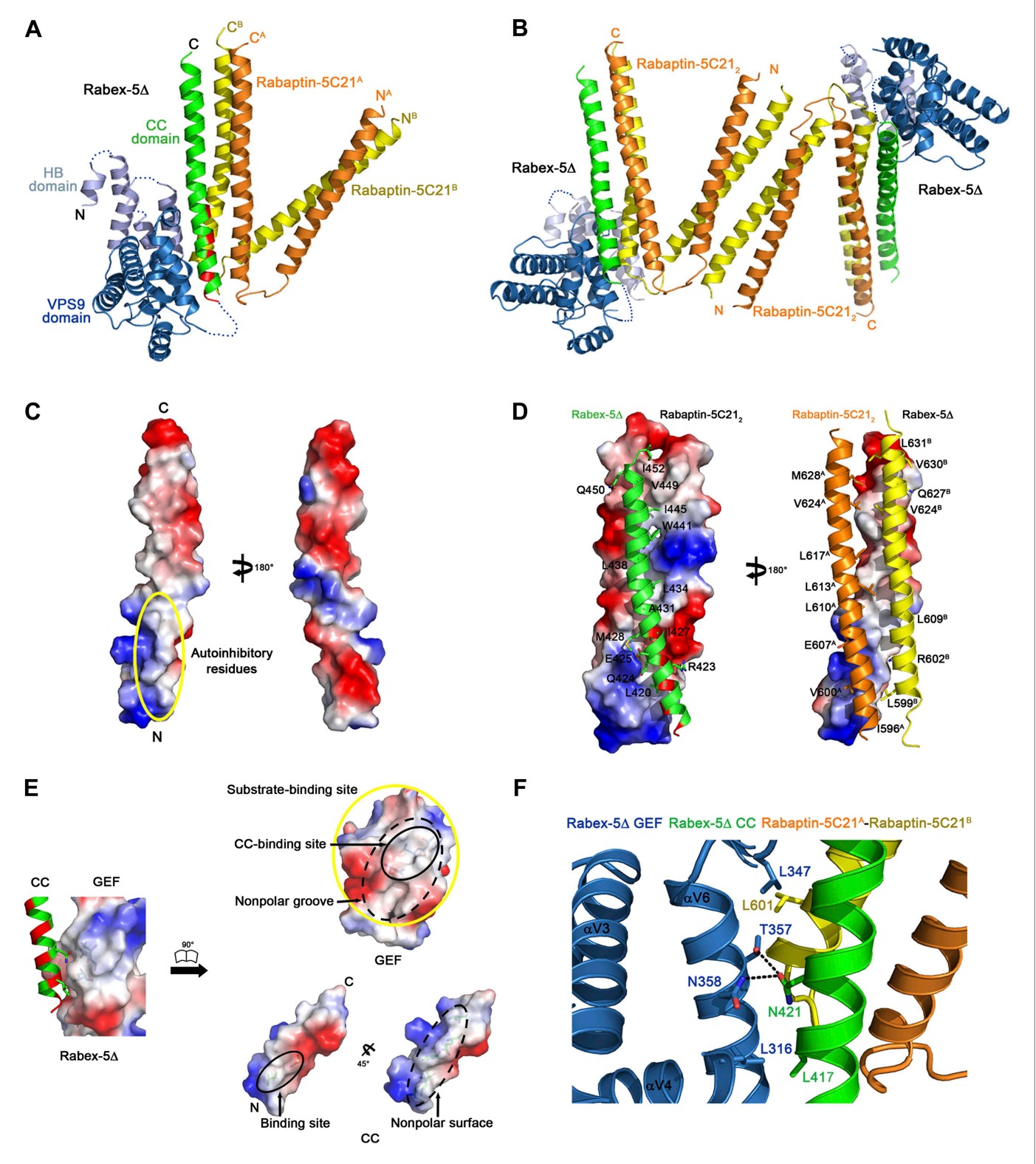

**Figure 1**. Crystal structure of the Rabex-5Δ-Rabaptin-5C21₂ complex. (**A**) A ribbon representation of the overall structure of the Rabex-5Δ-Rabaptin-5C21₂ complex. The HB and Vps9 domains of the Rabex-5 GEF domain are colored in light blue and dark blue, respectively, and the Rabex-5 CC domain (Rabex-5CC) in green. The two Rabaptin-5C21 are designated with superscripts A and B, and colored in orange and dark yellow, respectively. The disordered loops of the HB domain (residues 149, 161–162, 174, 190–204, and 220–230) and the linker between the GEF and CC domains (residues

*Figure 1. Continued on next page*

*Figure 1. Continued*

369–412Δ393-407) are indicated with dotted lines. The autoinhibitory residues of Rabex-5CC are marked in red. (**B**) The dimeric Rabex-5Δ-Rabaptin-5C21$_2$ complex. (**C**) An electrostatic surface representation of the amphipathic α-helix of Rabex-5CC. The autoinhibitory residues are located on the N-terminal of the nonpolar surface as indicated with a yellow circle. (**D**) Interactions between Rabex-5CC and Rabaptin-5C21$_2$. Left panel: Rabex-5CC is shown in ribbon representation and Rabaptin-5C21$_2$ in electrostatic surface representation. Right panel: Rabex-5CC is shown in electrostatic surface representation and Rabaptin-5C21$_2$ in ribbon representation. The interacting residues are shown with side chains. The detailed interactions is available in the *Figure 1—source data 1*. (**E**) An electrostatic surface representation of the interactions between the GEF and CC domains of Rabex-5Δ. (**F**) A close-up view of the interactions of the GEF domain with Rabex-5CC and Rabaptin-5C21. The interacting residues are shown with side chains and the hydrogen bonds are indicated with dotted lines.

The following source data and figure supplements are available for figure 1:

**Source data 1**. Interactions between Rabex-5CC and Rabaptin-5C21 in the Rabex-5Δ-Rabaptin-5C21$_2$ complex.

**Source data 2**. Interactions between Rabex-5CC and Rabaptin-5C21 in the Rabex-5CC-Rabaptin-5C21$_2$ complex.

**Figure supplement 1**. Schematic diagrams showing the domain organizations of Rabex-5 and Rabaptin-5.

**Figure supplement 2**. Trypsin digestion of the Rabex-5-Rabaptin-5C21 complex.

**Figure supplement 3**. Comparison of the Rabex-5 GEF domain in different structures.

**Figure supplement 4**. Crystal structure of Rabex-5CC.

**Figure supplement 5**. Crystal structure of the Rabex-5CC-Rabaptin-5C21$_2$ complex.

bundle, which is different from the V-shaped conformation in the R2Δ complex. Otherwise, two symmetry-related complexes also dimerize through the N-terminal regions of Rabaptin-5C21$_2$. Rabex-5CC also forms a long α-helix (52 Å) and packs in parallel with the C-terminal regions of Rabaptin-5C21$_2$ to form a tight three-helix bundle. The interactions between Rabex-5CC and Rabaptin-5C21$_2$ are essentially the same as those in the R2Δ complex. These results confirm that Rabex-5CC assumes a stable amphipathic α-helix and tends to bury its nonpolar surface via interactions with other proteins; Rabaptin-5C21 may adopt two different conformations and forms a stable homodimer; and the interactions between Rabex-5CC and Rabaptin-5C21$_2$ are conserved in different complexes.

## Structure of the Rab5-Rabex-5Δ-Rabaptin-5C21$_2$ complex

In the structure of the R2Δ complex, the substrate-binding site of Rabex-5 is partially occupied by Rabex-5CC and Rabaptin-5C21$_2$, implying that further conformational change(s) of the R2Δ complex will be required to bind Rab5. To investigate the molecular mechanism for Rab5 activation by the R2 complex, we solved the crystal structure of the Rab5-Rabex-5Δ-Rabaptin-5C21 (R3Δ) complex to 4.60 Å resolution (*Table 1*; *Figure 2A*, and *Figure 2—figure supplement 1*). The asymmetric unit contains two Rab5-Rabex-5Δ-Rabaptin-5C21$_2$ complexes related by a twofold non-crystallographic symmetry. The four Rabaptin-5C21 and two Rabex-5 are well defined in the electron density map; however, only one Rab5 is fairly defined while the other is poorly defined, indicating that the bound Rab5 has high flexibility which may explain the poor diffraction quality of the crystal. No nucleotide and/or metal ion are found at the active site of Rab5 and thus the bound Rab5 is in nucleotide-free form.

In the R3Δ complex, each Rabaptin-5C21 forms a long α-helix; two of them form a twisted linear two-helix bundle; and the two Rabaptin-5C21$_2$ dimerize through the middle regions (residues 590–600) (*Figure 2A* and *Figure 2—figure supplement 1A*). The conformation of Rabaptin-5C21$_2$ is similar to that in the Rabex-5CC-Rabaptin-5C21$_2$ complex and the previously reported GAT-Rabaptin-5C21$_2$ complex (*Zhu et al., 2004a*) but significantly different from that in the R2Δ complex (*Figure 2—figure supplement 2*). Similar to that in the R2Δ complex, Rabex5CC also forms a long α-helix and interacts with the C-terminal regions of Rabaptin-5C21$_2$ to form a three-helix bundle (*Figure 2A*). However, the orientations and positions of the Rabex-5 GEF domain and the N-terminal regions of Rabaptin-5C21$_2$ in relation to the three-helix bundle are dramatically different (*Figure 2B*). When the two complexes are superimposed based on the three-helix bundle, the N-terminal regions of Rabaptin-5C21$_2$ rotates

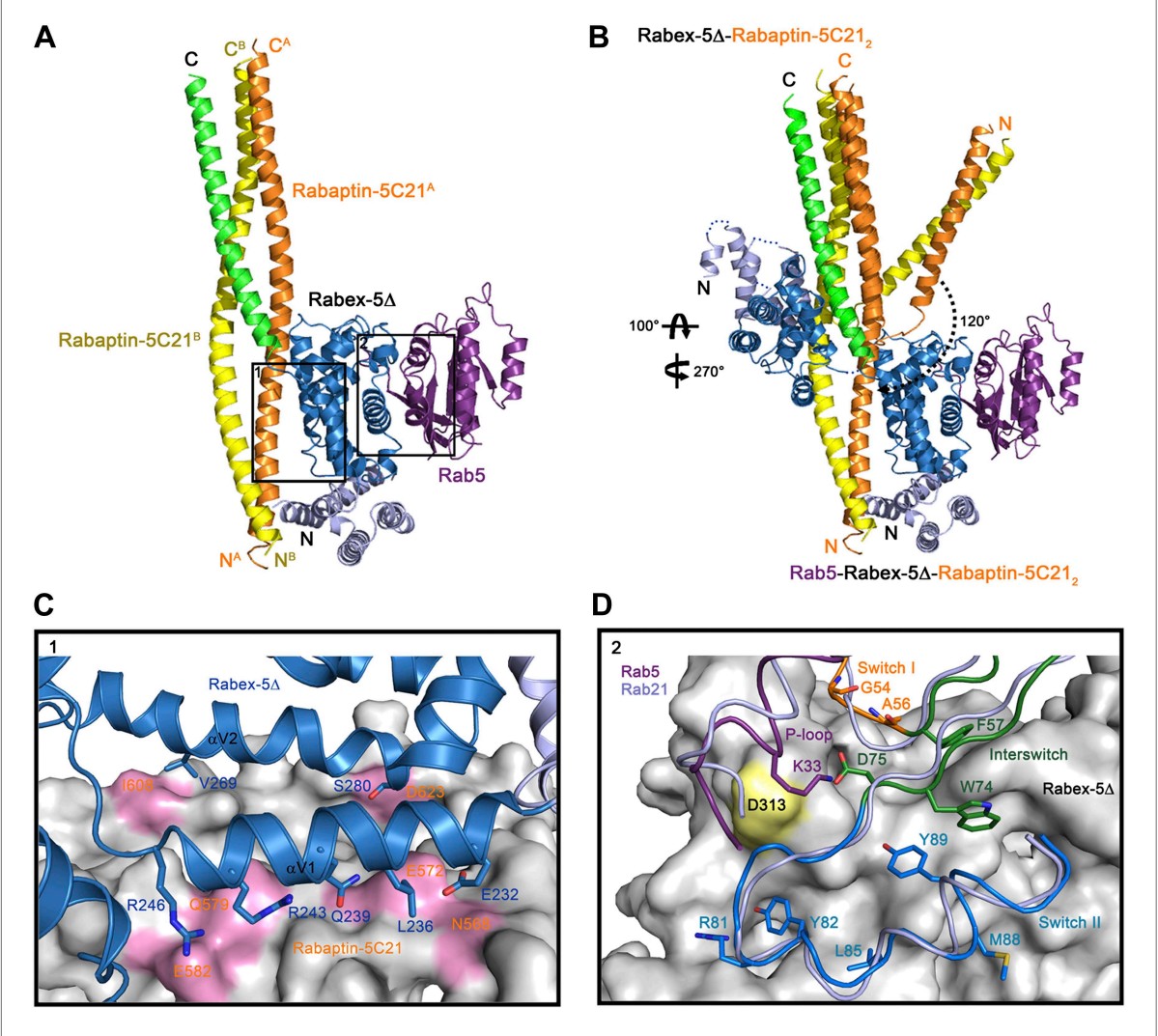

**Figure 2**. Crystal structure of the Rab5-Rabex-5Δ-Rabaptin-5C21$_2$ complex. (**A**) A ribbon representation of the overall structure of the Rab5-Rabex-5Δ-Rabaptin-5C21$_2$ complex. Only one Rab5-Rabex5Δ-Rabaptin5C21$_2$ complex in the asymmetric unit is shown with Rab5 in purple and Rabex-5 and Rabaptin-5C21 in the same colors as in *Figure 1A*. (**B**) Comparison of the Rab5-Rabex-5Δ-Rabaptin-5C21$_2$ complex and the Rabex-5Δ-Rabaptin-5C21$_2$ complex based on superposition of the three-helix bundle formed by Rabex-5CC and the C-terminal regions of Rabaptin-5C21$_2$. (**C**) Interactions between the Rabex-5 GEF domain and Rabaptin-5C21$_2$. The GEF domain is shown in ribbon representation in blue and the interacting residues are shown with side chains. Rabaptin-5C21$_2$ is shown in surface representation with the interacting residues colored in pink. (**D**) Interactions between Rab5 and the Rabex-5 GEF domain. Rab5 is shown in coil representation with the P-loop, switch I, switch II, and interswitch region colored in purple, orange, blue, and dark green, respectively. Several key residues are shown with side chains. The GEF domain is shown in surface representation with Asp313 colored in yellow. For comparison, Rab21 in its complex with the Rabex-5 GEF domain (*Delprato and Lambright, 2007*) is shown in coil representation in light blue.

The following figure supplements are available for figure 2:

**Figure supplement 1**. Crystal structure of the Rab5-Rabex-5Δ-Rabaptin-5C21$_2$ complex.

**Figure supplement 2**. Superposition of Rabaptin-5C21$_2$ in different complexes.

downwards by about 120° to transform from the V-shaped conformation to the linear conformation. Meanwhile, the GEF domain rotates by about 270° along the vertical axis and about 100° along the horizontal axis, and is dislodged from the three-helix bundle without any interaction. As a result, the substrate-binding site is completely exposed to the solvent for Rab5 binding.

The GEF domain of Rabex-5 interacts via a small portion of the opposite side of the substrate-binding site with a small portion of the N-terminal regions of Rabaptin-5C21$_2$ (*Figure 2A*). The

Rabex-5-Rabaptin-5C21 interaction involves only Glu232, Leu236, Gln239, Arg243, and Arg246 of αV1 and Val269 and Ser280 of αV2 of the GEF domain, and Asn568, Glu572, Gln579, and Glu582 of one Rabaptin-5C21 and Ile608 and Asp623 of the other, and the interaction interface buries a total of solvent accessible surface area of 1700 Å$^2$ (*Figure 2C*). The overall structure of the nucleotide-free Rab5 differs from the nucleotide-bound Rab5 in the conformations of the P-loop and the switch regions (*Merithew et al., 2001*; *Zhu et al., 2004b*). Nevertheless, the interactions between Rabex-5 and Rab5 are similar to those between Rabex-5 and Rab21, suggesting that Rabex-5 may activate Rab5 via a similar mechanism as for Rab21 (*Figure 2D*) (*Delprato and Lambright, 2007*; *Langemeyer et al., 2014*).

## Solution structure of the Rabex-5-Rabaptin-5C21$_2$ complex

In the R2Δ complex, Rabaptin-5C21$_2$ assumes a V-shaped conformation and the substrate-binding site of the Rabex-5 GEF domain is partially occupied by Rabex-5CC and Rabaptin-5C21$_2$ (*Figure 1*, *Figure 1—figure supplement 3C*). However, in the R3Δ complex, Rabaptin-5C21$_2$ adopts a linear conformation and the Rabex-5 GEF domain is displaced with a completely exposed substrate-binding site (*Figure 2A,B*). A modeling study shows that if Rabaptin-5C21$_2$ assumes the linear conformation in the R2Δ complex, the middle regions of Rabaptin-5C21$_2$ would have steric conflict with part of the Rabex-5 GEF domain (αV1, αV6, and αC helices), suggesting that the conformational change of Rabaptin-5C21$_2$ is essential for the complete exposure of the substrate-binding site of the GEF domain to bind Rab5. To investigate which conformation the R2 complex may assume in solution, we performed small angle X-ray scattering (SAXS) analyses of the R2, R2Δ, R3, and R3Δ complexes.

Our SAXS data show that the experimental P(r) distributions for R2Δ and R3Δ are similar to these of R2 and R3, respectively (*Figure 3—figure supplement 1A,B*), suggesting that deletion of the linker does not significantly affect the structures of R2Δ and R3Δ. The maximum paired-distance of the particle ($D_{max}$), the radius of gyration ($R_g$), and the Porod volume derived from the SAXS data together show that all of these complexes exist as dimers in solution, consistent with our biochemical and structural data (*Figure 3—source data 1*). The theoretical scattering curve calculated from the R2Δ model with the V-shaped conformation fits better with the experimental data for both R2 and R2Δ complexes (goodness of fit χ = 0.71 and 0.60, respectively) than that calculated from the R2Δ model with the linear conformation (goodness of fit χ = 0.74 and 0.65, respectively) (*Figure 3A,B*). In addition, although the theoretical and experimental P(r) distributions exhibit some differences, the theoretical P(r) distribution of the R2Δ model with the V-shaped conformation agrees better with the experimental P(r) distributions of both R2 and R2Δ complexes than that with the linear conformation (*Figure 3—figure supplement 1A,C*). As such, the solution structures of the R2 and R2Δ complexes can be best described as assuming mainly the V-shaped conformation. Nevertheless, it is plausible that the two segments of the V-shaped conformation and/or the component proteins of the complexes may bear some flexibility and adopt alternative conformation(s) in solution. Based on these results, we conclude that the R2 and R2Δ complexes assume mainly the V-shaped conformation as observed in the R2Δ structure with some flexibility in solution.

Similarly, the theoretical scattering curve calculated from the R3Δ model fits well with the experimental SAXS data for both R3 and R3Δ complexes (goodness of fit χ = 0.68 and 0.96, respectively) (*Figure 3C,D*), and the theoretical P(r) distribution of the R3Δ model is also in good agreement with the experimental P(r) distributions of both R3 and R3Δ complexes (*Figure 3—figure supplement 1B,D*). These results indicate that the R3 and R3Δ complexes assume mainly the linear conformation as observed in the R3Δ structure in solution.

## Functional analyses of the Rabex-5-Rabaptin-5 complex in Rab5 activation

To investigate the biological relevance of the R2Δ and R3Δ structures, we performed in vitro functional assays. We first measured the in vitro GEF activity of different Rabex-5 variants and mutants. Our kinetic data show that Rabex-5 containing the GEF and CC domains possesses a basal GEF activity ($0.93 \pm 0.03 \times 10^4$ M$^{-1} \cdot$s$^{-1}$); the Rabex-5 GEF domain alone exhibits a 3.2-fold higher activity ($2.93 \pm 0.06 \times 10^4$ M$^{-1} \cdot$s$^{-1}$); and the R2 complex exhibits a 3.3-fold higher activity ($3.07 \pm 0.08 \times 10^4$ M$^{-1} \cdot$s$^{-1}$) (*Figure 4A,B*). Moreover, the Rabex-5 mutants containing mutations on the nonpolar surface of Rabex-5CC exhibit relatively higher GEF activity (1.4–1.9 folds) compared with the wild-type Rabex-5 (*Figure 4—figure supplement 1A*). These results indicate that the GEF domain itself is constitutively

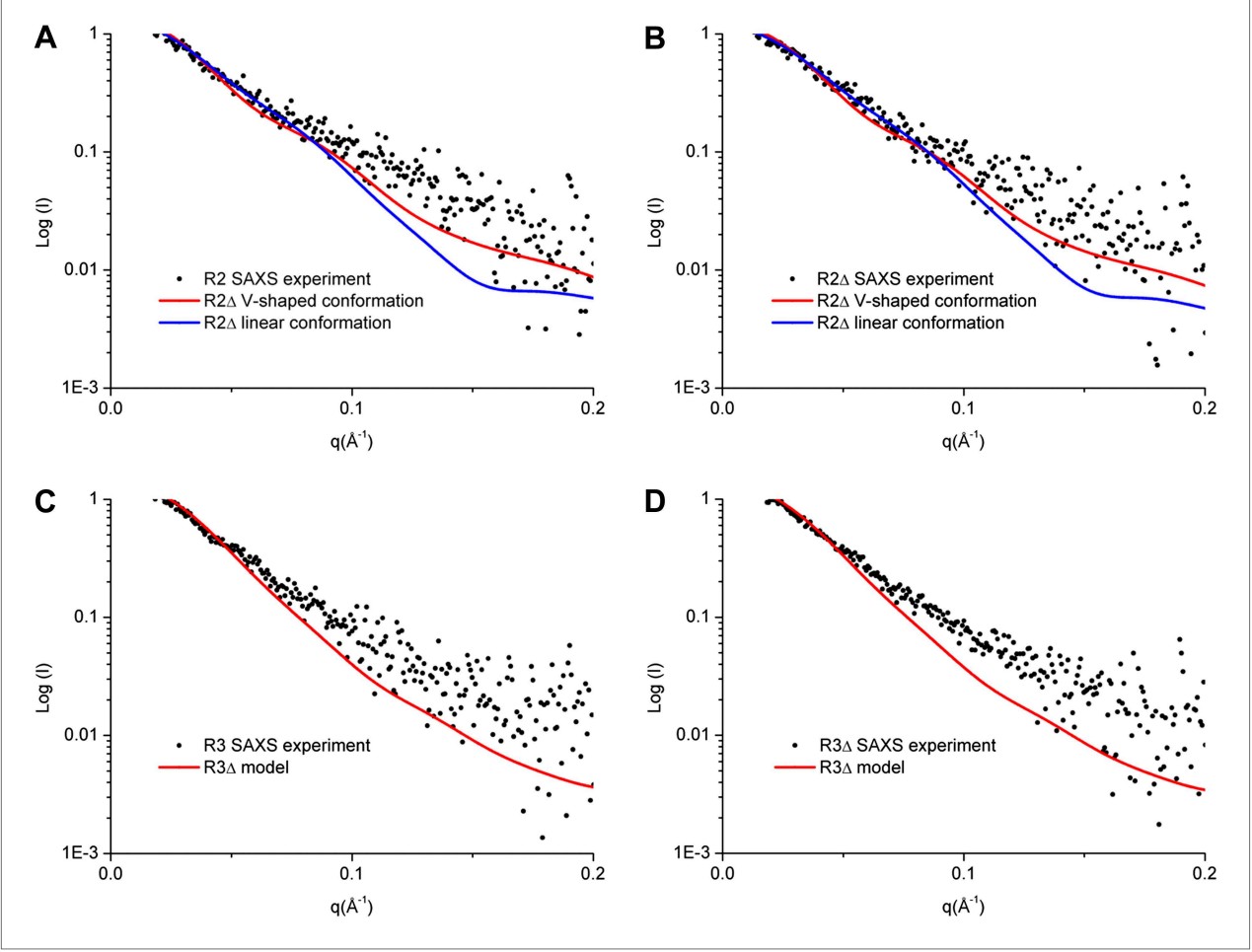

**Figure 3**. SAXS analyses of the R2 and R3 complexes. (**A** and **B**) Comparison of the experimental data with the theoretical scattering curves calculated from the structure models of R2Δ with the V-shaped conformation observed in the R2Δ structure and the linear conformation observed in the R3Δ structure for the R2 complex (**A**) and the R2Δ complex (**B**). (**C** and **D**) Comparison of the experimental data with the theoretical scattering curve calculated from the structure model of R3Δ observed in the R3Δ structure for the R3 complex (**C**) and the R3Δ complex (**D**). The observed and calculated values of $R_g$, $D_{max}$, and Porod volume are summarized in *Figure 3—source data 1*.

The following source data and figure supplements are available for figure 3:

**Source data 1**. SAXS analysis parameters.

**Figure supplement 1**. SAXS analyses of the R2 and R3 complexes.

active; the CC domain slightly autoinhibits the GEF activity; and the binding of Rabaptin-5C21 to Rabex-5CC relieves the autoinhibition, which are largely in agreement with the previous biochemical data (*Lippe et al., 2001*; *Delprato et al., 2004*; *Zhu et al., 2007*; *Langemeyer et al., 2014*). However, the magnitude of the autoinhibition by Rabex-5CC is smaller than that reported by *Delprato and Lambright (2007)*, which may be caused by differences of the assay systems, for examples, different Rab5 and Rabex-5 constructs, different concentration of the proteins, and different sensitivity of the instruments.

In the R2Δ complex, the linker between the Rabex-5 GEF and CC domains was removed to facilitate the crystallization. We then investigated whether the linker deletion has any effects on the functions of Rabex-5 and the R2 complex. Our kinetic data show that Rabex-5Δ possesses a slightly higher activity (1.9-fold) than Rabex-5 and the R2Δ complex exhibits a similar activity (0.9-fold) as the R2 complex (*Figure 4A,B*). Meanwhile, our in vitro GST pull-down assay results show that Rabex-5Δ can bind tightly to Rabaptin-5C21; however, deletion of either the N- or C-terminal half of Rabex-5CC disrupts

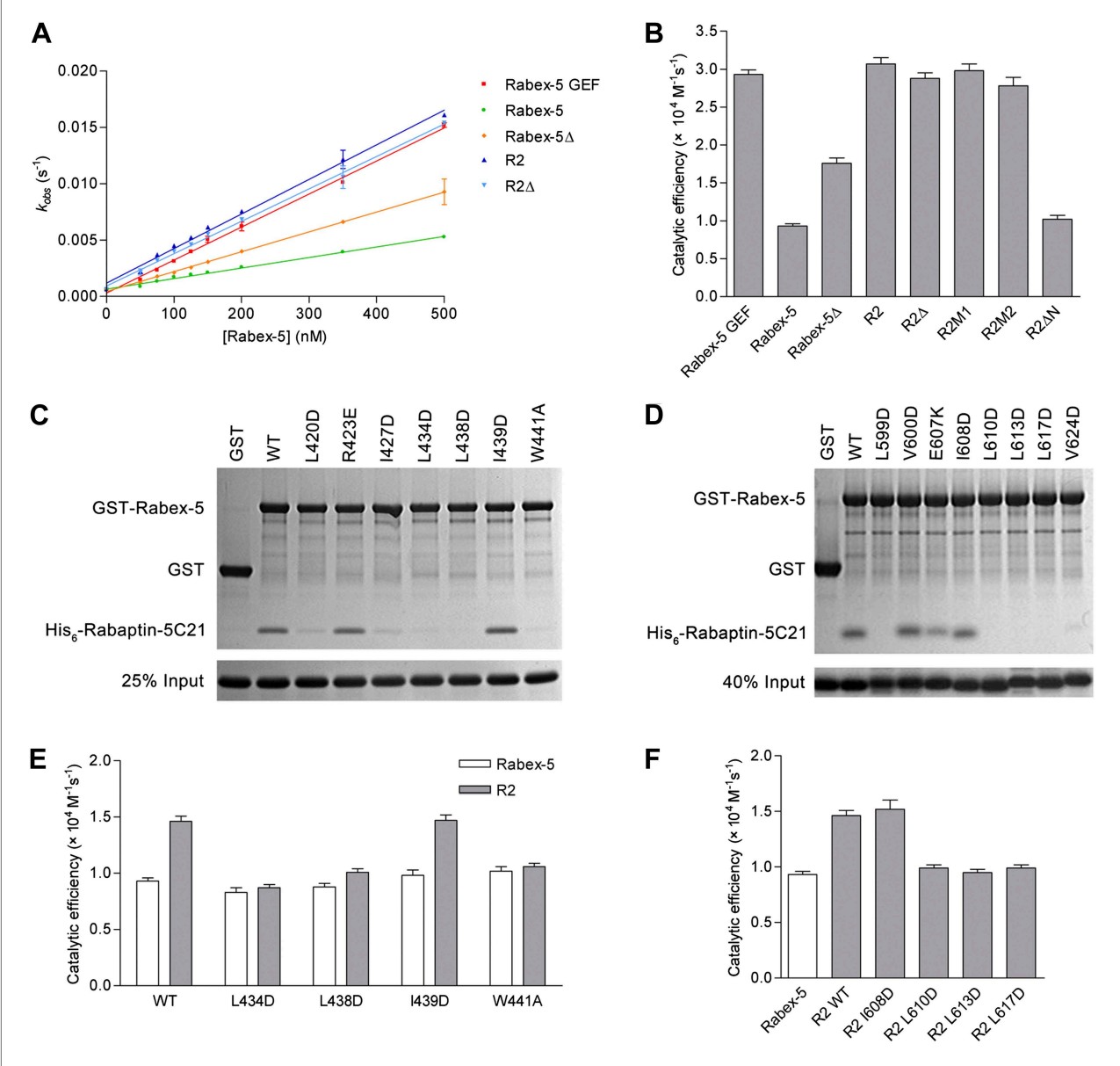

**Figure 4.** In vitro functional analyses of the Rabex-5-Rabaptin-5 complex. (**A**) GEF activity of Rabex-5 in different forms. Catalytic efficiency ($k_{cat}/K_m$) was obtained from the slope of a linear least-squares-fit of the $k_{obs}$ values against the concentrations of Rabex-5 from two independent measurements. (**B**) Histogram of the catalytic efficiencies of Rabex-5 variants alone and in complexes with different Rabaptin-5C21 mutants or variant. Values are means ± SEM of two independent measurements. R2M1: the R2 complex in which Rabaptin-5C21 contains a quadruple mutation N568A/E572A/Q579A/E582A; R2M2: the R2 complex in which Rabaptin-5C21 contains a double mutation I608A/D623A; R2ΔN: the R2 complex in which the N-terminal half of Rabaptin-5C21 (residues 552–592) is deleted. The complexes in (**A**) and (**B**) were co-expressed and co-purified. (**C**) GST pull-down assays for the interactions between the wild-type and mutant GST-Rabex-5 and the wild-type His$_6$-Rabaptin-5C21. The gel was stained by Coomassie blue. (**D**) GST pull-down assays for the interactions between the wild-type GST-Rabex-5 and the wild-type and mutant His$_6$-Rabaptin-5C21. (**E**) Histogram of the catalytic efficiencies of the wild-type and mutant Rabex-5 alone and in complexes with the wild-type Rabaptin-5C21. (**F**) Histogram of the catalytic efficiencies of the wild-type Rabex-5 alone and in complexes with the wild-type and mutant Rabaptin-5C21. For the assays in (**E**) and (**F**), Rabex-5 and Rabaptin-5C21 were expressed and purified separately and then mixed together in a 1:2 molar ratio overnight prior to the assay. Tables of the GEF activities are available in the *Figure 4—source data 1–3*.

The following source data and figure supplements are available for figure 4:

**Source data 1**. GEF activity of different Rabex-5 variants alone and in complexes with different Rabaptin-5C21 mutants or truncates.

*Figure 4. Continued on next page*

*Figure 4. Continued*

**Source data 2**. GEF activity of different Rabex-5 mutants alone and in complexes with wild-type Rabaptin-5C21.
**Source data 3**. GEF activity of wild-type Rabex-5 in complexes with different Rabaptin-5C21 mutants.
**Figure supplement 1**. Functional roles of the Rabex-5 CC domain (Rabex-5CC) in the autoinhibition of the Rabex-5 GEF activity.

the interaction (*Figure 4—figure supplement 1B*). These results indicate that the linker plays a minor role in the autoinhibition of the GEF activity but is not involved in the Rabex-5-Rabaptin-5C21 interaction, and the linker deletion does not affect the function of the R2 complex in the Rab5 activation.

In the R2Δ and Rabex-5CC-Rabaptin-5C21$_2$ complexes, the interactions between Rabex-5CC and Rabaptin-5C21 are well conserved. To validate the biological relevance of these interactions, we mutated several key residues of both Rabex-5CC and Rabaptin-5C21 at the interaction interface and analyzed their effects on the Rabex-5-Rabaptin-5 interaction. Our in vitro GST pull-down assay results show that mutations L434D, L438D, and W441A of Rabex-5CC, and mutations L599D, L610D, L613D, and L617D of Rabaptin-5C21 abolish the interaction, and mutations L420D and I427D of Rabex-5CC and mutation V624D of Rabaptin-5C21 substantially impair the interaction. In contrast, mutations R423E of Rabex-5CC and E607K of Rabaptin-5C21 have no significant effect on the interaction as these two residues form a salt bridge on the solvent-exposed surface and thus their mutations do not affect the hydrophobic core of the interaction. As negative controls, mutations I439D of Rabex-5CC and I608D of Rabaptin-5C21 have no effect on the interaction as these two residues are located on the polar surface of the three-helix bundle and are not involved in the interaction (*Figure 4C,D*). In agreement with the GST pull-down results, our kinetic data show that the GEF activity of the L434D, L438D, and W441A Rabex-5 mutants cannot be activated by Rabaptin-5C21, whereas that of the I439D Rabex-5 mutant can be potentiated by Rabaptin-5C21 (*Figure 4E*). Similarly, the L610D, L613D, and L617D Rabaptin-5C21 mutants cannot relieve the autoinhibition of Rabex-5; whereas the I608D Rabaptin-5C21 mutant can activate the GEF activity of Rabex-5 (*Figure 4F*). It is noteworthy that due to partial aggregation of Rabaptin-5C21, the GEF activity of the mixed R2 complex is only 1.6-fold higher compared with Rabex-5 which is weaker than the co-expressed and co-purified R2 complex (3.3-fold).

In the R3Δ complex, the Rabex-5 GEF domain interacts via a small surface opposite the substrate-binding site with a small portion of the N-terminal regions of Rabaptin-5C21$_2$ (*Figure 2C*). To explore the functional role of this interaction, we constructed two Rabaptin-5C21 mutants containing a quadruple mutation (N568A/E572A/Q579A/E582A) and a double mutation (I608A/D623A) of the residues on the interaction interface and detected their effects on the GEF activity of the R2 complex. Our kinetic data show that the GEF activity of these mutant complexes are unaffected (*Figure 4B*), indicating that this interaction is not essential for the activation of the Rabex-5 GEF activity. Intriguingly, when the N-terminal half of Rabaptin-5C21 (residues 552–592) was removed, the GEF activity of Rabex-5 could not be activated (*Figure 4B*), indicating that the full-length Rabaptin-5C21 is required for the function of the R2 complex in the Rab5 activation. Taken together, these data indicate that the structures of the R2Δ and R3Δ complexes are functionally relevant, and the interaction between Rabex-5 and Rabaptin-5 is important for the activation of the Rabex-5 GEF activity in Rab5 activation, which is in accord with the previous biochemical data (*Lippe et al., 2001*).

## Discussion

The previous biochemical and biological data showed that Rabex-5 functions together with Rabaptin-5 to activate Rab5 and then to promote the fusion of early endosomes in endocytosis (*Lippe et al., 2001*; *Delprato and Lambright, 2007*; *Zhu et al., 2007*). The GEF activity of Rabex-5 is autoinhibited by its CC domain and is activated by the binding of Rabaptin-5 via its C2-1 domain. However, the molecular mechanism is unknown. In this work, we determined the crystal structures of Rabex-5Δ in complex with Rabaptin-5C21 and in complex with Rabaptin-5C21 and Rab5, which are validated by biophysical and biochemical analyses.

Our structural data show that at the substrate-binding site of the Rabex-5 GEF domain, there is a surface groove composed of largely nonpolar residues. Rabex-5CC forms a stable amphipathic α-helix

that tends to bury its nonpolar surface via oligomerization or interaction with the C-terminal regions of Rabaptin-5C21$_2$. The nonpolar surface of Rabex-5CC has good chemical and geometrical complementarities with the nonpolar surface groove of the GEF domain and thus might be able to bind there to block the substrate binding and hence autoinhibit the GEF activity as proposed by *Delprato and Lambright (2007)*. Nonetheless, our structural and biochemical data show that although Rabex-5CC alone exists as a stable helix in both solution and crystal structure, it cannot form a stable complex with the GEF domain as shown by both GST pull-down assay and ITC analysis (data not shown). In addition, Rabex-5 and Rabex-5Δ cannot be crystallized alone and Rabex-5 can be easily proteolyzed in the linker region (*Figure 1—figure supplement 2*). These results suggest that the CC domain and the linker have high flexibility. Moreover, Rabex-5 itself has a basal GEF activity which is slightly weaker (about 1/3) than that of the constitutively active GEF domain (*Figure 4A,B*) but is not so weak compared with some other GEFs including DSS4 (a GEF for Ypt1p) (*Esters et al., 2001*) and MSS4 (a GEF for Rab8) (*Zhu et al., 2001*; *Itzen et al., 2006*). Mutations of the residues on the nonpolar surface of Rabex-5CC can enhance the GEF activity by 1.4–1.9 folds (*Figure 4—figure supplement 1A*). These results together indicate that the binding of Rabex5CC to the GEF domain is not tight, and Rabex-5 alone is not completely autoinhibited.

On the other hand, in the structure of the R2Δ complex, Rabaptin-5C21$_2$ forms 2 two-helix bundles with a V-shaped conformation, and Rabex-5CC interacts via the nonpolar surface with the C-terminal regions of Rabaptin-5C21$_2$ to form a tight three-helix bundle. Meanwhile, the GEF domain folds along the three-helix bundle with a partially occupied substrate-binding site, suggesting that further conformational change is required to completely expose the substrate-binding site for Rab5 binding and activation. Indeed, in the structure of the R3Δ complex, although Rabex-5CC still forms a tight three-helix bundle with the C-terminal regions of Rabaptin-5C21$_2$, Rabaptin-5C21$_2$ forms a linear two-helix bundle which is different from the V-shaped conformation but similar to that in the Rabex-5CC-Rabaptin-5C21$_2$ complex and the GAT-Rabaptin-5C21$_2$ complex (*Zhu et al., 2004a*). The GEF domain is dislodged from the three-helix bundle and interacts with the N-terminal regions of Rabaptin-5C21$_2$, and the substrate-binding site is completely exposed to bind Rab5. Meanwhile, our SAXS analysis results indicate that the R2 and R2Δ complexes mainly assume the V-shaped conformation as observed in the R2Δ structure but have some flexibility in solution, and our biochemical data show that the R2 and R2Δ complexes have full GEF activities as the constitutively active GEF domain and the N-terminal regions of Rabaptin-5C21$_2$ are required for the function of the R2 complex in the activation of Rab5.

Based on the structural and biological data in this work and those reported previously, we can propose the molecular mechanism for the regulation of the Rabex-5 GEF activity despite the lack of an intact Rabex-5 structure (*Figure 5*). In the free-form Rabex-5, Rabex-5CC may bind weakly via the nonpolar surface to the substrate-binding site of the GEF domain, leading to the blockage of the substrate-binding site and thus a weak autoinhibition of the GEF activity (*Figure 5*, State I). As the binding of Rabex-5CC to the GEF domain is not tight, it might assume alternative conformations. The linker between the GEF and CC domains might help to modulate the conformational flexibility and/or the relative conformations of the two domains and thus plays a minor role in the autoinhibition. One plausible alternative conformation of Rabex-5CC might be similar to that observed in the R2Δ structure with a largely exposed substrate-binding site of the GEF domain. As the interaction of Rabex-5CC with the GEF domain via the nonpolar surface is much tighter than that via the adjacent surface, Rabex-5CC might exist mainly in the autoinhibitory conformational state and partially in the alternative conformational state. This may explain why the free-form Rabex5 exhibits some basal GEF activity. Rabaptin-5C21 might bind to the C-terminal region of the nonpolar surface of Rabex-5CC in the autoinhibitory conformational state and induce conformational change of the α-helix to transform into the conformational state as observed in the R2Δ structure, or directly to the exposed nonpolar surface of Rabex-5CC in the alternative conformational state, leading to the relief of the autoinhibition and the release of the GEF activity (*Figure 5*, State II).

When the Rabex-5-Rabaptin-5 complex is recruited to the early endosomal membrane via the interaction of the C-terminal region of Rabaptin-5 with the GTP-bound Rab5, Rabex-5 can activate Rab5 locally in a very efficient way. The binding of the GDP-bound Rab5 to Rabex-5 induces further conformational changes of the Rabex-5-Rabaptin-5 complex such that Rabaptin-5C21$_2$ transforms from the V-shaped to the linear conformation, the GEF domain is dislodged from Rabex-5CC and the C-terminal regions of Rabaptin-5C21$_2$, and the substrate-binding site is completely exposed to the solvent to bind Rab5 as observed in the R3Δ structure (*Figure 5*, State III). In this conformational state,

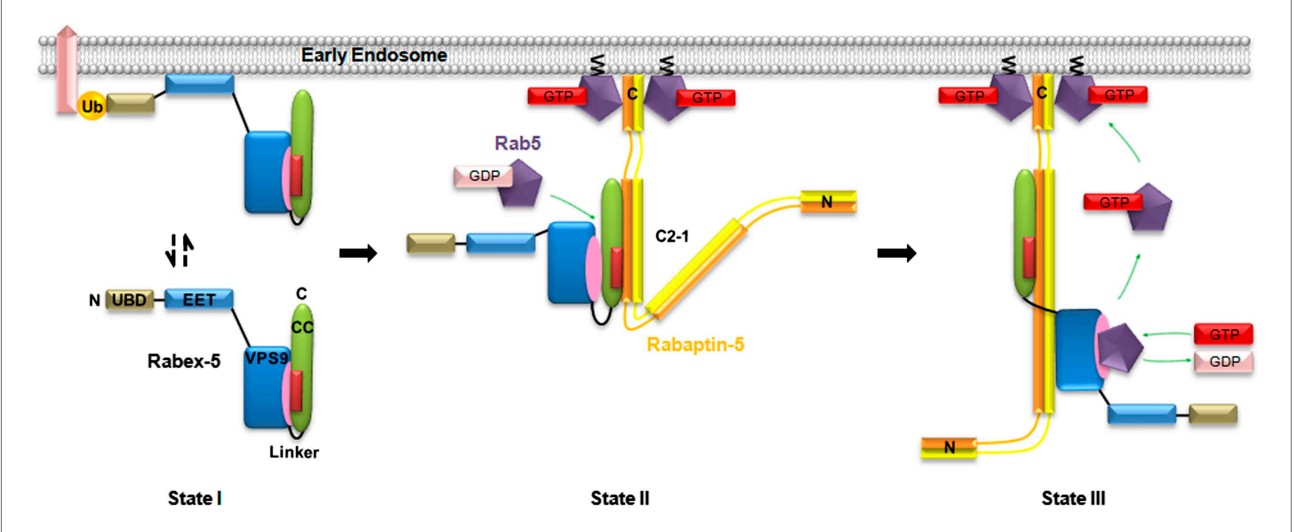

**Figure 5**. Molecular mechanism of the regulation of the Rabex-5 GEF activity. In the free-form Rabex-5, the CC domain binds weakly via the nonpolar surface to the substrate-binding site of the GEF domain, leading to occlusion of the substrate-binding site and thus a weak autoinhibition of the GEF activity (State I). The free-form Rabex-5 can directly target to the early endosomes to activate Rab5 at the basal level. In the cells, most Rabex-5 forms a binary complex with Rabaptin-5 which can be recruited to early endosomes via the binding of the C-terminal region of Rabaptin-5 to GTP-bound Rab5. The binding of Rabaptin-5C21 to Rabex-5 pulls Rabex-5CC away from the GEF domain to form a binary complex with the V-shaped conformation and a largely exposed substrate-binding site, leading to the relief of the autoinhibition (State II). The binding of Rab5 induces further conformational changes of the Rabex-5-Rabaptin-5 complex such that Rabaptin-5C21$_2$ transforms from the V-shaped to the linear conformation, and the substrate-binding site of the GEF domain is completely exposed to the solvent to bind and activate Rab5 (State III). The positive feedback loop among Rab5, its effector Rabaptin-5, and its GEF Rabex-5 can lead to a robust activation of Rab5, which then promotes the fusion of early endosomes efficiently.

the Rabex-5-Rabaptin-5 complex can facilitate the exchange of GDP- to GTP-bound Rab5. The not-so-tight interaction of the Rabex-5 GEF domain with Rabex-5CC and Rabaptin-5C21$_2$ as observed in the R2Δ structure allows the conformational changes easily during the substrate binding. The conformational changes of the Rabex-5-Rabaptin-5 complex induced by the binding of Rab5 leading to the full activation of the Rabex-5 GEF activity may provide another leverage to ensure the high substrate specificity.

In the context of the early endosomal membrane, Rabex-5 can directly target to the early endosomes either through the interaction of the N-terminal ubiquitin binding domain (UBD) with the ubiquitinated cargoes or through the early endosomal targeting domain (EET) (*Zhu et al., 2007*; *Mattera and Bonifacino, 2008*). In this case, as the GEF activity of Rabex-5 is autoinhibited by Rabex-5CC, Rabex-5 can only activate Rab5 at the basal level. In the cells, most Rabex-5 forms a stable complex with Rabaptin-5 which can be recruited to early endosomes via the binding of Rabaptin-5 through the C-terminal region to GTP-bound Rab5 (*Lippe et al., 2001*; *Zhu et al., 2010*). In this case, Rabaptin-5 not only assists the recruitment of Rabex-5 to the early endosomal membrane, but also activates the GEF activity of Rabex-5. The positive feedback loop among Rab5, its effector Rabaptin-5, and its GEF Rabex-5 can lead to a robust activation of Rab5, which then promotes the fusion of early endosomes efficiently.

As a scaffold protein, Rabaptin-5 comprises several coiled-coil regions which can mediate interactions with different proteins to exert different functions. In addition to acting as the effector of Rab5 to function in the fusion of early endosomes, Rabaptin-5 can interact with Rab4 via the N-terminal region and thus may serve as the effector of Rab4 to function in the endocytic recyling process (*Vitale et al., 1998*). As the C2-1 domain of Rabaptin-5 can interact with the CC domain of Rabex5 and the GAT domain of GGA1, and Rabaptin-5 is the effector of Rab5 and GGA1 is the effector of Arf1, this dual interaction might mediate the crosstalk between Rab and Arf GTPases to promote the tethering and fusion of early endosomes and trans-Golgi network-derived vesicles (*Mattera et al., 2003*; *Zhu et al., 2004a*; *Kawasaki et al., 2005*).

In the GAT-Rabaptin-5C21$_2$ structure, Rabaptin-5C21$_2$ assumes a linear conformation, the N-terminal regions of Rabaptin-5C21$_2$ bind one GAT, and two symmetry-related Rabaptin-5C21$_2$ dimerize through the C-terminal regions (*Zhu et al., 2004a*), which are the binding site for Rabex-5CC. Interestingly, Rabaptin-5C21$_2$ assumes a linear conformation in the Rabex-5CC-Rabaptin-5C21$_2$ structure but a

V-shaped conformation in the R2Δ structure, and in both structures, the C-terminal regions of Rabaptin-5C21$_2$ bind one Rabex-5CC and two symmetry-related Rabaptin-5C21$_2$ dimerize through the N-terminal regions which are the binding site for GAT. Moreover, in the R3Δ structure, Rabaptin-5C21$_2$ also assumes a linear conformation but two Rabaptin-5C21$_2$ dimerize through the middle regions (residues 590–600) between the GAT-binding site and the Rabex-5CC-binding site. As the GAT-binding site is unoccupied, Rabaptin-5 may bind Rabex-5 and GGA1 simultaneously. These results demonstrate that Rabaptin-5C21 always forms a dimer which can assume either a V-shaped or a linear conformation and can bind other proteins via different regions. In addition, the dimeric Rabaptin-5C21$_2$ prefers to further dimerize to form a dimer of dimers via the regions that are not involved in the interactions with other proteins. In the context of full-length Rabaptin-5, it is possible that Rabaptin-5C21$_2$ might exist in either the V-shaped or the linear conformation depending on the functional state. The dimerization of Rabaptin-5C21$_2$ might not only avoid exposure of the hydrophobic surface and minimize the overall energy of the protein in aqueous environment, but also play some functional roles in the tethering and fusion of early endosomes and/or with other vesicles.

## Materials and methods

### Cloning, expression, and purification of proteins

The cDNAs corresponding to the Rabex-5 CC domain (residues 409–455, Rabex-5CC), the Rabex-5 GEF domain (residues 132–392), the Rabex-5 GEF and CC domains (residues 132–455, Rabex-5), the Rabaptin-5 C2-1 domain (residues 552–642, Rabaptin-5C21), and Rab5 (residues 15–184) were all amplified by PCR from the cDNA library of human brain cells. The Rabex-5CC and Rabex-5 GEF ORFs were cloned into the pET-28a plasmid (Novagen, Germany) with a His$_6$ tag inserted at the N-terminus. The Rab5 ORF was cloned into a modified pET-28a plasmid (Novagen) with a His$_6$-sumo tag inserted at the N-terminus. The Rabex-5CC and Rabaptin-5C21 ORFs, and the Rabex-5 and Rabaptin-5C21 ORFs were cloned into the pET-Duet1 plasmid (Novagen) with a His$_6$ tag inserted at the N-terminus of Rabex-5CC and Rabex-5, respectively. The Rabex-5Δ variants containing different deletion forms of the linker between the GEF and CC domains were generated using the Takara MutanBEST Mutagenesis kit (TakaRa Biotechnology, Japan). The Rabex-5 and Rabaptin-5C21 mutants containing point mutations were generated using the QuikChange Site-Directed Mutagenesis kit (Agilent Technologies, Santa Clara, CA).

All recombinant proteins were expressed in *E. coli* BL21 (DE3) Codon-Plus strain (Novagen). The transformed cells were grown at 37°C in LB medium containing 0.05 mg/ml ampicillin or kanamycin until OD$_{600}$ reached 0.8, and then induced with 0.25 mM IPTG at 16°C for 24 hr. All the proteins were purified by affinity chromatography using a Ni-NTA column (Qiagen, Germany) and gel filtration chromatography using a Superdex 200 16/60 column (GE Healthcare, Sweden) in a buffer containing 20 mM Tris–HCl, pH 8.0, 150 mM NaCl, and 1 mM PMSF. The resultant samples were of >95% purity as evaluated by SDS-PAGE.

### Trypsin digestion analysis

A trypsin stock solution (2.5 mg/ml) was diluted to $10^{-1}$ to $10^{-6}$ times. The Rabex-5-Rabaptin-5C21 complex (1 mg/ml) was mixed with the trypsin solution of different concentrations. The digestion reaction proceeded for 30 min at 4°C and 16°C, respectively, and was then stopped by addition of 10 μg/ml aprotinin to inhibit the activity of trypsin. The reaction mixture was loaded onto Ni-NTA beads, and both the beads and the flow-through were analyzed by SDS-PAGE with Coomassie blue staining and Western blot with anti-His antibody (1:3000, TIANGEN, China).

### In vitro GST pull-down assay

For in vitro GST pull-down assay, the Rabaptin-5C21 ORF was cloned into the pET-3E-His plasmid (Novagen) with an N-terminal His$_6$ tag, and the Rabex-5 ORF into the pGEX 6P-1 plasmid (GE Healthcare) with an N-terminal GST tag. His$_6$-Rabaptin-5C21 was purified by Ni-NTA affinity chromatography and GST-Rabex-5 by glutathione sepharose beads (GE Healthcare). 20 μg GST-Rabex-5 immobilized onto the glutathione sepharose beads were incubated with 100 μg His$_6$-Rabaptin-5C21 at 4°C for 2 hr. The beads were analyzed by SDS-PAGE with Coomassie blue staining.

### Nucleotide exchange assay

The Rabex-5 GEF activity for Rab5 was determined using the method described previously (*Delprato et al., 2004*). Briefly, Rab5 was mixed with 20-fold excess fluorescent 2′(3′)-bis-O-(N-methylanthraniloyl)-GDP

(mantGDP, Invitrogen, Carlsbad, CA). The mixture was incubated for 2 hr and the free mantGDP was removed by gel filtration using a HiTrap De-salting column (GE Healthcare). The mantGDP-bound Rab5 was diluted to 500 nM in a buffer containing 20 mM Tris–HCl (pH 8.0), 150 mM NaCl, and 2 mM $MgCl_2$. Nucleotide exchange reaction was initiated by addition of GTP to a final concentration of 1 mM and varied concentrations (50–500 nM) of Rabex-5 or Rabex-5-Rabaptin-5C21. Dissociation of mantGDP was monitored by measuring the decrease of fluorescence. Samples were excited at 360 nm and the emission was monitored at 440 nm. Fluorescence data were recorded using a Varian Cary Eclipse spectrofluorimeter (Agilent Technologies). Observed pseudo first-order exchange rate constant ($k_{obs}$) was obtained by a nonlinear least-squares-fit of the data at each concentration of Rabex-5 to the exponential equation

$$I(t) \ = \ (I_0 - I_\infty) \exp(-k_{obs}t) + I_\infty$$

where I(t) is the emission intensity at time t, $I_0$ the initial emission intensity, and $I_\infty$ the final emission intensity. Catalytic efficiency ($k_{cat}/K_m$) was obtained from the slope of a linear least-squares-fit of the $k_{obs}$ values to the linear equation

$$k_{obs} \ = \ (k_{cat}/K_m)[Rabex-5] + k_{intr}$$

where $k_{intr}$ is the intrinsic nucleotide exchange rate in the absence of Rabex-5. The intrinsic exchange rate ($k_{intr}$) of Rab5 is measured to be $0.00064 \pm 0.00002$ $s^{-1}$.

## Crystallization, data collection, and structure determination

Crystallization was performed using the hanging drop vapor diffusion method at 16°C by mixing equal volumes (1.0 µl) of protein solution (20 mg/ml) and reservoir solution. Crystals of the R2Δ complex were grown from drops consisting of the reservoir solution of 2.0 M $NaH_2PO_4$/$K_2HPO_4$ (pH 7.0) and 0.05% n-octyl-β-D-galactopyranoside. Crystals of Rabex-5CC were grown from drops consisting of the reservoir solution of 0.10 M NaAc (pH 5.4), 17.5% MPD, and 2% PEG4000. Crystals of the Rabex-5CC-Rabaptin-5C21 complex were grown from drops consisting of the reservoir solution of 0.15 M $MgAc_2$ and 20% PEG3350. Crystals of the R3Δ complex were grown from drops consisting of the reservoir solution of 1.0 M $NaH_2PO_4$/$K_2HPO_4$ (pH 5.0). All of the diffraction data were collected at −175°C at beamline 17U of Shanghai Synchrotron Radiation Facility, and processed with HKL2000 (*Otwinowski and Minor, 1997*).

The structure of the R2Δ complex was solved using the molecular replacement (MR) method as implemented in Phenix (*Adams et al., 2010*) with the structure of the Rabex-5 GEF domain (PDB code 1TXU) (*Delprato et al., 2004*) as the search model. The structure of the Rabex-5CC-Rabaptin-5C21 complex was solved by MR with the structure of the Rabaptin-5 C2-1 domain (PDB code 1X79) (*Zhu et al., 2004a*) as the search model. The structure of Rabex-5CC was solved by MR with the structure of Rabex-5CC in its complex with Rabaptin-5C21 as the search model. The structure of the R3Δ complex was solved by MR with the structure of the Rabex-5 GEF-Rab21 complex (PDB code 2OT3) (*Delprato and Lambright, 2007*) as the search model.

Structure refinement was carried out using Phenix (*Adams et al., 2010*), Refmac5 (*Murshudov et al., 1997*), and CNS (*Brunger, 2007*), and model building using Coot (*Emsley and Cowtan, 2004*). Due to the low resolution of the diffraction data, the structure models of Rab5, Rabex-5, and Rabaptin-5C21 in the R3Δ complex were refined as rigid bodies with deformable elastic network and group B-factor restraints (*Schroder et al., 2010*) and thus the side-chain orientations in this complex are somewhat uncertain. Stereochemistry of the structure models was analyzed using Procheck (*Laskowski et al., 1993*). Structural analyses were carried out using programs in CCP4 (*Winn et al., 2011*) and the PISA server (*Krissinel and Henrick, 2007*). All structure figures were generated using PyMOL (http://www. pymol.org). The statistics of the structure refinement and final structure models are summarized in *Table 1*.

## Small angle X-ray scattering (SAXS) analysis

Protein samples were concentrated to 5 mg/ml in 20 mM Tris–HCl (pH 8.0) and 150 mM NaCl. Solution scattering experiments were performed at 293 K on a SAXSess $mc^2$ platform (Anton Paar, Austria) equipped with a sealed tube source and a CMOS diode array detector. The SAXS data were collected with 2 hr exposure time in 1-hr frame to ensure absence of radiation damage during the course of the experiment. The SAXS data for the buffer were recorded for background subtraction. Inverse Fourier transformation was performed with the GIFT program in the PCG software package (Anton Paar). The maximum paired-distance ($D_{max}$) value was extrapolated from the P(r) distribution. The radius of gyration ($R_g$) and the Porod volume were calculated using PRIMUS (*Konarev et al.,*

*2003*) at the low angle region (q × $R_g$ ≤1). The theoretical P(r) distribution and $R_g$ value for each structure model were calculated using PTRAJ from the AMBER12 package (*Case et al., 2012*). The structure model was assessed against the corresponding solution scattering data using CRYSOL from the ATSAS software (*Svergun et al., 1995*) with constant subtraction. The his$_6$-tag and disordered residues were built back into the crystal structures using PyMOL (http://www.pymol.org), and the yielded structure models were optimized using Xplor-NIH (*Schwieters et al., 2006*) for optimal packing. For each crystal structure, a total of 800 structure models were generated with Monte Carlo simulated annealing while fixing the coordinates of the atoms observed in the crystal structure.

## Accession codes

The crystal structures of Rabex-5CC, the Rabex-5CC-Rabaptin-5C21$_2$ complex, the Rabex-5Δ-Rabaptin-5C21$_2$ complex, and the Rab5-Rabex-5Δ-Rabaptin-5C21$_2$ complex have been deposited with the Protein Data Bank under accession codes 4N3X, 4N3Y, 4N3Z, and 4Q9U, respectively.

## Acknowledgements

We thank the staff members at beamline 17U of Shanghai Synchrotron Radiation Facility (SSRF), China for technical support in diffraction data collection. We are grateful to Lan Bao, Xu-qiao Chen, Haixiang Shi, Fang Yu, and Jing Feng for their valuable contributions to the research work. This work was supported by grants from the National Natural Science Foundation of China (31230017) and the Ministry of Science and Technology of China (2011CB966301 and 2011CB911102).

## Additional information

### Funding

| Funder | Grant reference number | Author |
|---|---|---|
| National Natural Science Foundation of China | 31230017 | Zhe Zhang, Tianlong Zhang, Shanshan Wang, Jianping Ding |
| Ministry of Science and Technology of the People's Republic of China | 2011CB966301 | Zhe Zhang, Tianlong Zhang, Shanshan Wang, Jianping Ding |
| Ministry of Science and Technology of the People's Republic of China | 2011CB911102 | Zhe Zhang, Tianlong Zhang, Shanshan Wang, Jianping Ding |

The funders had no role in study design, data collection and interpretation, or the decision to submit the work for publication.

### Author contributions

ZZ, Carried out the cloning, protein purification, crystallization, structure determination and analyses, biophysical and biochemical studies, and drafted and revised the manuscript; TZ, Participated in the structure determination and analyses; SW, Participated in the cloning and protein purification; ZG, Participated in the design, analysis and discussion of the SAXS study; CT, Participated in the design, analysis and discussion of the SAXS study, and revised the manuscript; JC, Participated in the design, analysis and discussion of the biochemical studies; JD, Conceived the study, participated in the designs and data analyses of all experiments, and wrote and revised the manuscript

## Additional files

### Major dataset

The following datasets were generated:

| Author(s) | Year | Dataset title | Dataset ID and/or URL | Database, license, and accessibility information |
|---|---|---|---|---|
| Zhang Z, Zhang T, Ding J | 2013 | Crystal structure of Rabex-5 CC domain | 4N3X; http://www.rcsb.org/pdb/search/structidSearch.do?structureId=4N3X | Publicly available at the RCSB Protein Data Bank (http://www.rcsb.org/pdb/) |

| Zhang Z, Zhang T, Ding J | 2013 | Crystal structure of Rabex-5CC and Rabaptin-5C21 complex | 4N3Y; http://www.rcsb.org/pdb/search/structidSearch.do?structureId=4N3Y | Publicly available at the RCSB Protein Data Bank (http://www.rcsb.org/pdb/) |
|---|---|---|---|---|
| Zhang Z, Zhang T, Ding J | 2013 | Crystal structure of Rabex-5delta and Rabaptin-5C21 complex | 4N3Z; http://www.rcsb.org/pdb/search/structidSearch.do?structureId=4N3Z | Publicly available at the RCSB Protein Data Bank (http://www.rcsb.org/pdb/) |
| Zhang Z, Zhang T, Ding J | 2014 | Crystal structure of the Rab5, Rabex-5delta and Rabaptin-5C21 complex | 4Q9U; http://www.rcsb.org/pdb/search/structidSearch.do?structureId=4Q9U | Publicly available at the RCSB Protein Data Bank (http://www.rcsb.org/pdb/) |

The following previously published datasets were used:

| Author(s) | Year | Dataset title | Dataset ID and/or URL | Database, license, and accessibility information |
|---|---|---|---|---|
| Delprato A, Merithew E, Lambright DG | 2004 | Crystal Structure of the Vps9 Domain of Rabex-5 | 1TXU; http://www.rcsb.org/pdb/explore/explore.do;jsessionid=696B7B648E345F4C30FD67B32F1FA992?structureId=1TXU | Publicly available at the RCSB Protein Data Bank (http://www.rcsb.org/pdb/) |
| Delprato A, Lambright DG | 2007 | Crystal structure of rabex-5 VPS9 domain in complex with nucleotide free RAB21 | 2OT3; http://www.rcsb.org/pdb/explore/explore.do;jsessionid=696B7B648E345F4C30FD67B32F1FA992?structureId=2OT3 | Publicly available at the RCSB Protein Data Bank (http://www.rcsb.org/pdb/) |
| Zhu G, Zhai P, He X, Wakeham N, Rodgers K, Li G, Tang J, Zhang XC | 2004 | Crystal structure of human GGA1 GAT domain complexed with the GAT-binding domain of Rabaptin5 | 1X79; http://www.rcsb.org/pdb/explore/explore.do;jsessionid=696B7B648E345F4C30FD67B32F1FA992?structureId=1X79 | Publicly available at the RCSB Protein Data Bank (http://www.rcsb.org/pdb/) |

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
