## [Decision Letter]

Thank you for sending your work entitled “Molecular mechanism of the activation of Rab5 by Rabex-5 and Rabaptin-5 in endocytosis” for consideration at *eLife*. Your article has been favorably evaluated by a Senior editor and three reviewers, one of whom, Suzanne Pfeffer, is a member of our Board of Reviewing Editors. The Reviewing editor and the other reviewers discussed their comments before we reached this decision, and the Reviewing editor has assembled the following comments to help you prepare a revised submission.

Zhang et al. report crystal structures of: the Rabex-5 GEF and coiled coil domains with an internal flexible loop deleted (Rabex-5delta) in complex with the coiled coil Rabex-5 binding regions of Rabaptin-5 (Rabaptin-5C212); the coiled coil region of Rabex-5 alone; the latter in complex with Rabex-5 binding region of Rabaptin-5; and a low resolution ternary complex of Rabex-5delta:Rabaptin-5C212 bound to nucleotide free Rab5. Collectively, these structures represent an important contribution to the field by defining the mode of interaction between the coiled coil regions of Rabex-5 and Rabaptin-5, identifying alternative conformational states and interactions with the substrate site that may have a functional role, and providing an overview of the quaternary organization of the active complex with a substrate Rab. The structural information is complemented by kinetic, binding, and cell morphologic analyses of the constructs used for the crystallographic studies as well as the effects of various site specific mutants. Finally, the authors also investigated whether the alternative conformations are compatible with SAXS experiments in solution.

1) In the kinetic assays, how was the intrinsic exchange rate for Rab5 taken into account? It should be roughly 0.0005 s-1; however, the fitted lines in Figure 3 all appear to extrapolate to zero. Also, the concentration range analyzed is somewhat narrow (50-150 nM or 3 fold). A broader range may be necessary to measure both high and low catalytic efficiencies. For example, low catalytic efficiencies may be poorly estimated in the range appropriate for high catalytic efficiencies and a higher/broader range may be required to distinguish catalyzed from intrinsic exchange. Since the longest Rabex-5 construct is used as a reference for the mutants but has the weakest activity, it is important to measure its activity in an appropriate concentration range.

2A) The statement on that “the P(r) functions of the R2 and R2delta complexes exhibit a saddle-shaped peak especially for R2, suggesting that both R2 and R2delta complexes assume more than one conformation in solution” is confusing. How can the presence of multiple conformations be inferred from features in the P(r) distribution without having the P(r) distribution of the relevant individual conformations for comparison? The presence of multiple peaks in the P(r) is not unexpected for proteins with multiple domains or proteins that form oligomeric complexes (see, for example, Putnam et al., 2007, Quart. Rev. Biophys 40, 191-285). Before ascribing significance to specific features in the P(r) distributions for the experimental data, it would seem important to compare them with the theoretical P(r) distributions for the R2 and R2delta structures.

2B) If a constant was not subtracted when fitting the data in Crysol, does constant subtraction improve the fits such that one of the conformations can adequately describe the data? A better analysis (e.g. with Bunch) would include the his-tag and any other disordered regions. A more complicated model with two conformations may fit the data better but that doesn't prove that the model is correct. It is important to account for missing elements before concluding that a more complicated model is required.

2C) Dmax is not the only measure of molecular size and its estimation can be confounded by tags or disordered regions. Is the Porod volume (should be approximately half the molecular mass) also consistent with a dimer? Is the Guinier constant Rg larger than expected for monomers? Does it correspond to the calculated Rg for either the linear or V-shaped structures?

2D) The description of the methods used for SAXS analysis is incomplete. E.g., what program was used to calculate P(r)? What procedure was used to determine Dmax? Was constant subtraction used when fitting the data in Crysol?

3) The contribution of rabaptin-5 to membrane targeting of Rabex-5 is not fully considered. Previous work from the Zerial lab has demonstrated synergy between Rabex-5 and rabaptin-5. A simple explanation for this would be that rabaptin-5 binding both promotes or stabilizes membrane association of Rabex-5 and stimulates its activity by relieving auto-inhibition. Such a model would fit with the modest activation of core GEF activity seen on rabaptin binding. To test this idea the localization of Rabex-5 point mutants with compromised rabaptin interaction would be informative. This should be done at low expression levels, but is not essential for publication.

4) Figure 3 the text claims in vivo functional assays were performed. This is the weakest area of the manuscript. The images in Figure 3 and Figure 3–figure supplement 3 show Rab5 localization in BHK cells transfected with Rabex-5 constructs in the presence of the endogenous Rabex-5. 2. The experiments in 3G depend upon equal expression of Rab5 as well as the various Rabex constructs introduced and a stable Rab5 cell line would be most appropriate here. In any case, co-staining for the constructs and quantitation of expression levels would add confidence to the findings reported. Quantitations on endosome size are mentioned in the Figure 3–figure supplement 3 legend text but it is not clear how such numbers could be obtained from the data presented here. The staining could be Golgi from the images presented. First, the images don't show any obvious endosome staining and also require another marker. Second, these experiments need to be done in the absence of endogenous Rabex-5. Finally, some form of functional assay for Rab5 dependent endocytic transport should be performed, for example growth factor or transferrin uptake. The best way to fix this would be to rescue cells depleted of Rabex-5 and monitor endocytosis after transfection with various rescue constructs.

Other comments:

1) Figure 1 is very busy and confusing at first glance. It is not immediately obvious that the green helix in Figure 1 is part of Rabex-5 because the labeling overlaps the structure. The presentation and labeling should be improved.

2) The text should make it clearer that nucleotide-free Rab5 was present in the crystal structure. This is only mentioned later nd described in a way that makes it appear surprising Rab5 is in nucleotide-free form. Taking the work of [6] and [18] together one would have expected Rab5 and Rab21 to be activated by the same mechanism and to show the same interactions to Rabex-5 when in nucleotide-free form.

3) Figure 2 should show only one complex and not both copies in the overall structure for clarity. The text indicates that one Rab5 is not well defined, so this could be shown in the supplement rather than the main figure.

4) The Discussion and model should better explain how the complex might interact with and sit at a membrane surface given the known interactions of Rabex-5 via its UIM and of rabaptin with GGAs and Rabs. 3. Also, the coiled coil of Rabaptin seems to exist as an either bent form or elongated form in solution and can form a 3-helix bundle with RABEX5 to open the enzyme. Not clear is whether this coiled coil is freely available to behave that way within the context of the full length Rabaptin-5 protein and this needs to be discussed and included in Figure 5. Also not clearly specified is why Lambright saw more dramatic inhibition and relief thereof, by the Rabaptin domain (please add this to the Discussion).

5) The Title is not a correct description of the work. The authors uncover details of the way in which rabaptin binding relieves auto inhibition of Rabex-5. It does not provide the mechanism of Rab5 activation in detail or look at endocytosis.

6) The statement that the “the substrate-binding site is largely exposed to the solvent and accessible by the substrate” appears to be contradicted by the previous sentence, which states that “the binding sites for switch II and a small portion of the interswitch region of Rab21 are occupied by the three-helix bundle” and by Figure 1 and Figure 1—figure supplement 4 in which the substrate binding site appears to be blocked by the three helix bundle.

7) Abstract: “Rabaptin-5C212” should be defined on first usage. In particular, clarify what the “2” after “Rabaptin-5C21” signifies.

8) Several abbreviations (e.g., R2, R2delta, R3, etc) are missing from the list. The authors should decrease jargon wherever possible as there are many acronyms here. Thus, please define R2 complex? R3 complex? Constructs used in 3G? It is hard for the reader.

9) Finally, because the assignment and orientation of side chains shown in Figure 2 are determined primarily by the molecular replacement models and only a little bit by the data, please add a qualifying note about the uncertainty of the side chain orientations due to the low resolution of the structure.

---

## [Author Response]

*1) In the kinetic assays, how was the intrinsic exchange rate for Rab5 taken into account? It should be roughly 0.0005 s-1; however, the fitted lines in*
Figure 3
*all appear to extrapolate to zero. Also, the concentration range analyzed is somewhat narrow (50-150 nM or 3 fold). A broader range may be necessary to measure both high and low catalytic efficiencies. For example, low catalytic efficiencies may be poorly estimated in the range appropriate for high catalytic efficiencies and a higher/broader range may be required to distinguish catalyzed from intrinsic exchange. Since the longest Rabex-5 construct is used as a reference for the mutants but has the weakest activity, it is important to measure its activity in an appropriate concentration range*.

We thank the reviewers for the constructive suggestion. In the revision, to broaden the concentration range of Rabex-5, we have measured the exchange rates of different Rabex-5 constructs at three additional concentrations of 200 nM, 350 nM, and 500 nM, and fitted the entire kinetic data to the linear equation “*k*_obs_ = (*k*_cat_/*K*_m_)[Rabex-5]+*k*_intr_” to derive the catalytic efficiencies (*k*_cat_*/K*_m_) from the slope of the line. The new results are provided in Figure 4, Figure 4—figure supplement 1, and [Supplementary-material SD4-data SD5-data SD6-data]. The newly obtained catalytic efficiencies are consistent with our previous results. In addition, we have measured the intrinsic exchange rate (*k*_intr_) of Rab5 and as expected by the reviewers, the intrinsic exchange rate (*k*_intr_) of Rab5 is 0.00064 ± 0.00002 s^-1^. In the revision, we have elaborated the nucleotide exchange assay method in the Materials and methods section (Nucleotide exchange assay) accordingly.

*2A) The statement on that “the P(r) functions of the R2 and R2delta complexes exhibit a saddle-shaped peak especially for R2, suggesting that both R2 and R2delta complexes assume more than one conformation in solution“ is confusing. How can the presence of multiple conformations be inferred from features in the P(r) distribution without having the P(r) distribution of the relevant individual conformations for comparison? The presence of multiple peaks in the P(r) is not unexpected for proteins with multiple domains or proteins that form oligomeric complexes (see, for example, Putnam et al., 2007, Quart. Rev. Biophys 40, 191-285). Before ascribing significance to specific features in the P(r) distributions for the experimental data, it would seem important to compare them with the theoretical P(r) distributions for the R2 and R2delta structures*.

*2B) If a constant was not subtracted when fitting the data in Crysol, does constant subtraction improve the fits such that one of the conformations can adequately describe the data? A better analysis (e.g. with Bunch) would include the his-tag and any other disordered regions. A more complicated model with two conformations may fit the data better but that doesn't prove that the model is correct. It is important to account for missing elements before concluding that a more complicated model is required*.

We thank the reviewers for pointing out the misstatement about the saddle-shaped peak. As these two comments are related, we respond to them together. In the revision, we have re-analyzed the SAXS data using the methods suggested by the reviewers and revised the text accordingly. As suggested by the reviewers, we have added the his_6_-tag and the disordered residues back into the crystal structures, and the yielded structure models were optimized and assessed against the experimental SAXS data. We have also performed constant subtraction during the fitting.

The experimental P(r) distribution for the R2Δ mutant is similar to that of R2 (Figure 3—figure supplement 1), suggesting that deletion of the linker does not affect the structure of R2Δ in solution significantly.

As shown in Figure 3, the theoretical scattering curve calculated from the R2Δ model with the V-shaped conformation fits better with the experimental SAXS data for both R2 and R2Δ complexes (goodness of fit χ= 0.71 and 0.60, respectively) than that calculated from the R2Δ model with the linear conformation (goodness of fit χ= 0.74 and 0.65, respectively). In addition, we compared the experimental P(r) distributions for the R2 and R2Δ complexes with the theoretical P(r) distributions of the R2Δ models with the V-shaped conformation and the linear conformation (Figure 3—figure supplement 1). Although the theoretical and experimental P(r) distributions exhibit some differences, the theoretical P(r) distribution of the R2Δ model with the V-shaped conformation agrees better with the experimental P(r) distributions of both R2 and R2Δ complexes than that with the linear conformation. As such, the solution structures of R2Δ and R2 can be best described as assuming mainly the V-shaped conformation. Nevertheless, it is plausible that the two segments of the V-shaped conformation and/or the component proteins of the complex may bear some flexibility and adopt alternative conformation(s) in solution. As pointed out by the reviewers, a more complicated model with two or more conformations may fit the experimental data better but that does not prove the correctness of the model. Based on these data as well as the values of the radius of gyration (R_g_), the maximum paired-distance (D_max_), and the Porod volume, we conclude that the R2 and R2Δ complexes assume mainly the V-shaped conformation with some flexibility in solution.

2C) Dmax is not the only measure of molecular size and its estimation can be confounded by tags or disordered regions. Is the Porod volume (should be approximately half the molecular mass) also consistent with a dimer? Is the Guinier constant Rg larger than expected for monomers? Does it correspond to the calculated Rg for either the linear or V-shaped structures?

Based on the structure model of R2Δ, we also computed the theoretical radius of gyration (R_g_) and maximum paired-distance (D_max_), and compared them with the experimental values. Based on the assessments, the R2Δ dimer with the V-shaped conformation would be best to describe the solution structures of R2 and R2Δ. In addition, the experimental Porod volumes for both R2 and R2Δ(210 nm^3^ and 200 nm^3^, respectively) are in agreement with the molecular masses of R2 and R2Δ in dimeric form (118 kDa and 114 kDa, respectively). The observed and calculated values of R_g_, D_max_, and Porod volume are now summarized in [Supplementary-material SD3-data].

2D) The description of the methods used for SAXS analysis is incomplete. E.g., what program was used to calculate P(r)? What procedure was used to determine Dmax? Was constant subtraction used when fitting the data in Crysol?

We have now elaborated the SAXS analysis procedures in the Materials and methods section (Small angle X-ray scattering analysis).

*3) The contribution of rabaptin-5 to membrane targeting of Rabex-5 is not fully considered. Previous work from the Zerial lab has demonstrated synergy between Rabex-5 and rabaptin-5. A simple explanation for this would be that rabaptin-5 binding both promotes or stabilizes membrane association of Rabex-5 and stimulates its activity by relieving auto-inhibition. Such a model would fit with the modest activation of core GEF activity seen on rabaptin binding. To test this idea the localization of Rabex-5 point mutants with compromised rabaptin interaction would be informative. This should be done at low expression levels, but is not essential for publication*.

We concur with the reviewers that the Rabaptin-5 binding to Rabex-5 both promotes or stabilizes the membrane association of Rabex-5 and stimulates its activity by relieving the autoinhibition. To test this idea, as suggested by the reviewers, we have observed the colocalization of wild-type or mutant Rabaptin-5 and Rabex-5 in HEK 293 cells co-transfected with GFP-Rab5, Myc-Rabaptin-5, and Flag-Rabex-5. As shown in Figure 5, we showed that both Rabex-5 (residues 132–455) and Rabex-5Δ (residues 132–455Δ393–407) colocalized well with Rabaptin-5 to the Rab5-positive early endosomes. However, in the cells co-transfected the Rabex-5 GEF domain with Rabaptin-5 or the Rabex-5 mutant containing a triple mutation L434D/L438D/W441A (Rabex-5 3M) with the Rabaptin-5 mutant containing a triple mutation L610D/L613D/L617D (Rabaptin-5 3M), although Rabaptin-5 can still bind to Rab5-positive early endosomes, the Rabex-5 GEF domain or the Rabex-5 mutant cannot be recruited to the early endosomes due to the disruption of the Rabex-5-Rabaptin-5 interaction. These results indicate that the interaction between Rabex-5 and Rabaptin-5 is important not only for the activation of the Rabex-5 GEF activity but also for the recruitment of Rabex-5 to the early endosomal membrane, which is in accord with the previous work from the Zerial lab and the expectation of the reviewers.

*4)*
Figure 3
*the text claims in vivo functional assays were performed. This is the weakest area of the manuscript. The images in*
Figure 3
*and Figure 3–figure supplement 3 show Rab5 localization in BHK cells transfected with Rabex-5 constructs in the presence of the endogenous Rabex-5. 2. The experiments in 3G depend upon equal expression of Rab5 as well as the various Rabex constructs introduced and a stable Rab5 cell line would be most appropriate here. In any case, co-staining for the constructs and quantitation of expression levels would add confidence to the findings reported. Quantitations on endosome size are mentioned in the Figure 3–figure supplement 3 legend text but it is not clear how such numbers could be obtained from the data presented here. The staining could be Golgi from the images presented. First, the images don't show any obvious endosome staining and also require another marker. Second, these experiments need to be done in the absence of endogenous Rabex-5. Finally, some form of functional assay for Rab5 dependent endocytic transport should be performed, for example growth factor or transferrin uptake. The best way to fix this would be to rescue cells depleted of Rabex-5 and monitor endocytosis after transfection with various rescue constructs*.

We thank the reviewers for this criticism and the constructive suggestions. As the Rab5 localization and function in BHK 21 cells were not obvious in our previous experiments, we obtained another GFP-Rab5 plasmid from Dr. Lan Bao’s laboratory in our institute which has been used in their researches and has shown to have good early endosomal localization, and re-performed the in vivo functional assays in HEK 293 cells.

We concur with the reviewers that our cell biological analyses might depend upon equal expression of Rab5 as well as the various Rabaptin-5 and Rabex-5 constructs introduced, and thus a stable Rab5 cell line would be most appropriate. However, we do not possess a stable Rab5 cell line from available resources. Thus, in the cell staining analyses, we co-transfected the cells with equal amounts of GFP-Rab5, Myc-Rabaptin-5, and Flag-Rabex-5 plasmids and used immunoblotting assays to examine the expression levels of the three proteins. Three independent experiments showed similar expression levels of these proteins (Figure 5C), indicating that the analysis method works properly.

According to the reviewers’ suggestion, we have co-stained all of the three proteins to detect their expression and colocalization when analyzing the size of early endosomes (Figure 5A). To confirm that the GFP-Rab5-marked vesicles are early endosomes, we co-stained the early endosomal marker EEA1. The representative images showed that the GFP-Rab5-marked vesicles were well labeled by EEA1 (Figure 5–figure supplement 1), and thus we still used the GFP-Rab5 signal to detect the size of early endosomes. Quantification of the endosome size followed the method used in literature (54). Specifically, three independent experiments were performed and the diameter of 100 largest GFP-Rab5-positive vesicles in 50∼60 cells was measured for each group. Statistical analyses were performed using PRISM (GraphPad Software Inc) with the two-tailed and unpaired Student’s *t*-test. The graph in Figure 5B shows the mean ± SEM.

We also concur with the reviewers that it would be better to deplete the endogenous Rabex-5 in our experiments. In the revision, we tried to use siRNA to deplete the endogenous Rabex-5. Unfortunately, the best siRNA among the three pairs we designed could only knock down the expression of the endogenous Rabex-5 by 55% (data not shown). Considering that the cotransfection efficiency with three (Rab5, Rabaptin-5, and Rabex-5) or four (+EGFR) plasmids is very low, this method appears to be very hard to work. In addition, for the cell morphological analyses, we have to mark the siRNA with a fluorescence label to monitor its transfection, which inevitably increases the difficulty to distinguish the siRNA signal from the fluorescence signals of the co-transfected proteins. Most of all, our results have shown that the cells co-transfected with Rabex-5 and Rabaptin-5 can enlarge the diameter of the early endosomes to 2.94 ± 0.09 μm which is significantly larger than that in the cells co-transfected with the vectors (0.97 ± 0.04 μm), suggesting that the influence of the endogenous Rabex-5 is limited and does not affect our analyses and conclusions. Thus, we remain to perform the assay in the presence of the endogenous Rabex-5.

For the functional analysis of Rab5 dependent endocytic transport, as reported in the literature, the activation of Rab5 does not affect the rate of EGF-stimulated EGFR internalization but may regulate the colocalization of internalized EGF/EGFR and Rab5 (Dinneen and Ceresa, 2004b). However, the activation of Rab5 facilitates the ligand-independent EGFR internalization (Dinneen and Ceresa, 2004a). In addition, although Rab5 can accelerate the transferrin internalization, there is no obvious difference between the wild-type and constitutively active mutant (Q79L) Rab5 (Stenmark et al, 1994). Therefore, to analyze whether the activation of Rab5 by the Rabex-5-Rabaptin-5 complex has any effect on the endocytic transport, we have performed both cell biological and biochemical experiments to detect the internalization of the unliganded EGFRs in HEK 293 cells (which do not express native EGFR) co-transfected with EGFR, Rab5, Rabex-5, and Rabaptin-5 (data not shown). Consistent with our cell biological results shown in Figure 5, the new results also showed that Rabex-5 and Rabaptin-5 can enlarge the size of early endosomes and disruption of the Rabex-5-Rabaptin-5 interaction decreases the size of early endosomes accordingly. We also observed that EGFR can colocalize well with Rab5 in each case, suggesting that the endocytic trafficking of EGFR is correlated to the function of Rab5. However, we did not observe notable difference of the EGFR internalization in the cells co-expressing the vectors and different forms of Rabex-5 and Rabaptin-5 in both the cell morphological analysis of the distribution of EGFRs and the biochemical analysis of the surface EGFRs by surface biotinylation assay (Chen et al, 2012). These results suggest that under our assay conditions, the activation of Rab5 by the Rabex-5-Rabaptin-5 complex has little effect on the internalization of EGFR but is essential for the proper function of Rab5 in the downstream processes, such as the fusion of early endosomes. Due to the indirect relevance of these results to the main topic of this paper, we decided not to include these results in the revised text.

Chen XQ, Wang B, Wu C, Pan J, Yuan B, Su YY et al (2012) Endosome-mediated retrograde axonal transport of P2X3 receptor signals in primary sensory neurons. *Cell Research*
**22:** 677-696.

Dinneen JL, Ceresa BP (2004a) Continual expression of Rab5(Q79L) causes a ligand-independent EGFR internalization and diminishes EGFR activity. *Traffic*
**5:** 606-615.

Dinneen JL, Ceresa BP (2004b) Expression of dominant negative rab5 in HeLa cells regulates endocytic trafficking distal from the plasma membrane. *Experimental Cell Research*
**294:** 509-522.

Stenmark H, Parton RG, Steele-Mortimer O, Lutcke A, Gruenberg J, Zerial M (1994) Inhibition of rab5 GTPase activity stimulates membrane fusion in endocytosis. *The EMBO Journal*
**13:** 1287-1296.

Zhu H, Zhu G, Liu J, Liang Z, Zhang XC, Li G (2007) Rabaptin-5-independent membrane targeting and Rab5 activation by Rabex-5 in the cell. *Molecular Biology of the Cell*
**18:** 4119-4128.

Other comments:

*1)*
Figure 1
*is very busy and confusing at first glance. It is not immediately obvious that the green helix in*
Figure 1
*is part of Rabex-5 because the labeling overlaps the structure. The presentation and labeling should be improved*.

As suggested by the reviewers, we have removed the labels of the secondary structures in Figure 1 and simply labeled the HB, Vps9, and CC domains of Rabex-5Δ with the same colors of the structure models. For clarity, we have also labeled the N- and C-terminals of Rabex-5Δ in black.

*2) The text should make it clearer that nucleotide-free Rab5 was present in the crystal structure. This is only mentioned later nd described in a way that makes it appear surprising Rab5 is in nucleotide-free form. Taking the work of*
[6]
*and [18] together one would have expected Rab5 and Rab21 to be activated by the same mechanism and to show the same interactions to Rabex-5 when in nucleotide-free form*.

As suggested by the reviewers, we have clarified in the text that the Rab5 in the Rab5-Rabex-5Δ-Rabaptin-5C21_2_ complex is a nucleotide-free form as follows: “No nucleotide and/or metal ion are found at the active site of Rab5 and thus the bound Rab5 is in nucleotide-free form.”

We have also cited the works of [6] and [18] when discussing the similar interactions of Rab5 and Rab21 with Rabex-5 in the text as follows: “Nevertheless, the interactions between Rabex-5 and Rab5 are similar to those between Rabex-5 and Rab21, suggesting that Rabex-5 may activate Rab5 via a similar mechanism as for Rab21 (Figure 2) (6; 18).”

*3)*
Figure 2
*should show only one complex and not both copies in the overall structure for clarity. The text indicates that one Rab5 is not well defined, so this could be shown in the supplement rather than the main figure*.

As suggested by the reviewers, we have now shown only one Rab5-Rabex-5Δ-Rabaptin-5C21_2_ complex in the overall structure in Figure 2 but both copies in Figure 2—figure supplement 1.

*4) The Discussion and model should better explain how the complex might interact with and sit at a membrane surface given the known interactions of Rabex-5 via its UIM and of rabaptin with GGAs and Rabs. 3. Also, the coiled coil of Rabaptin seems to exist as an either bent form or elongated form in solution and can form a 3-helix bundle with RABEX5 to open the enzyme. Not clear is whether this coiled coil is freely available to behave that way within the context of the full length Rabaptin-5 protein and this needs to be discussed and included in*
Figure 5*. Also not clearly specified is why Lambright saw more dramatic inhibition and relief thereof, by the Rabaptin domain (please add this to the Discussion)*.

As suggested by the reviewers, we have re-generated the schematic model to show the activation mechanism of Rabex-5 by Rabaptin-5 in the context of the early endosomal membrane localization of Rabex-5 and Rabaptin-5 in Figure 6 and discussed the model in more details in the text as follows: “In the context of the early endosomal membrane, Rabex-5 can directly target to the early endosomes either through the interaction of the N-terminal ubiquitin binding domain (UBD) with the ubiquitinated cargoes or through the early endosomal targeting domain (EET) (25; 54). In this case, as the GEF activity of Rabex-5 is autoinhibited by Rabex-5CC, Rabex-5 can only activate Rab5 at the basal level. In the cells, most Rabex-5 forms a stable complex with Rabaptin-5 which can be recruited to early endosomes via the binding of Rabaptin-5 through the C-terminal region to GTP-bound Rab5 (22; 53). In this case, Rabaptin-5 not only assists the recruitment of Rabex-5 to the early endosomal membrane, but also activates the GEF activity of Rabex-5. The positive feedback loop among Rab5, its effector Rabaptin-5, and its GEF Rabex-5 can lead to a robust activation of Rab5, which then promotes the fusion of early endosomes efficiently.”

As a scaffold protein, Rabaptin5 comprises several coiled-coil regions that mediate interactions with different proteins to exert different functions. Rabaptin-5C21 always forms a dimer, which can assume either a V-shaped conformation or a linear conformation and can bind other proteins via different regions. In addition, it prefers to further dimerize to form a dimer of dimers via the regions that are not involved in the interactions with other proteins. In the previous version of the manuscript, we did not discuss these issues due to space limitation. As suggested by the reviewers, in the revision, we have added a brief discussion on the oligomerization and conformational flexibility of Rabaptin-5 within the context of the full-length protein as follows: “In the GAT-Rabaptin-5C21_2_ structure, Rabaptin-5C21_2_ assumes a linear conformation, the N-terminal regions of Rabaptin-5C21_2_ bind one GAT, and two symmetry-related Rabaptin-5C21_2_ dimerize through the C-terminal regions (51) which are the binding site for Rabex-5CC. Interestingly, Rabaptin-5C21_2_ assumes a linear conformation in the Rabex-5CC-Rabaptin-5C21_2_ structure but a V-shaped conformation in the R2Δ structure, and in both structures, the C-terminal regions of Rabaptin-5C21_2_ bind one Rabex-5CC and two symmetry-related Rabaptin-5C21_2_ dimerize through the N-terminal regions which are the binding site for GAT. Moreover, in the R3Δ structure, Rabaptin-5C21_2_ also assumes a linear conformation but two Rabaptin-5C21_2_ dimerize through the middle regions (residues 590-600) between the GAT-binding site and the Rabex-5CC-binding site. As the GAT-binding site is unoccupied, Rabaptin5 may bind Rabex-5 and GGA1 simultaneously. These results demonstrate that Rabaptin-5C21 always forms a dimer which can assume either a V-shaped or a linear conformation and can bind other proteins via different regions. In addition, the dimeric Rabaptin-5C21_2_ prefers to further dimerize to form a dimer of dimers via the regions that are not involved in the interactions with other proteins. In the context of full-length Rabaptin-5, it is possible that Rabaptin-5C21_2_ might exist in either the V-shaped or the linear conformation depending on the functional state. The dimerization of Rabaptin-5C21_2_ might not only avoid exposure of the hydrophobic surface and minimize the overall energy of the protein in aqueous environment, but also play some functional roles in the tethering and fusion of early endosomes and/or with other vesicles.”

We think that the discrepancy between Lambright et al. and ours on the magnitude of the autoinhibition and the relief by the Rabaptin-5C21 binding might be caused by differences of the assay systems, for example, different Rab5 and Rabex-5 constructs, different concentration of the proteins, and different sensitivity of the instruments. In the revision, we have added these conjectures as follows: “However, the magnitude of the autoinhibition by Rabex-5CC is smaller than that reported by Delprato et al (6) which may be caused by differences of the assay systems, for examples, different Rab5 and Rabex-5 constructs, different concentration of the proteins, and different sensitivity of the instruments.”

*5) The Title is not a correct description of the work. The authors uncover details of the way in which rabaptin binding relieves auto inhibition of Rabex-5. It does not provide the mechanism of Rab5 activation in detail or look at endocytosis*.

We have changed the title to “Molecular mechanism of the activation of the Rabex-5 GEF activity by Rabaptin-5 in endocytosis”.

*6) The statement that the “the substrate-binding site is largely exposed to the solvent and accessible by the substrate” appears to be contradicted by the previous sentence, which states that ”the binding sites for switch II and a small portion of the interswitch region of Rab21 are occupied by the three-helix bundle” and by*
Figure 1
*and*
Figure 1—figure supplement 4
*in which the substrate binding site appears to be blocked by the three helix bundle*.

We thank the reviewers for the kindly reminder. In the Rabex-5Δ-Rabaptin-5C21_2_ complex, the substrate-binding site of Rabex-5 is partially occupied by the three-helix bundle. The interaction interface buries a total of solvent accessible surface area of 1040 Å^2^ which is much smaller than that between Rabex-5 and Rab21 (2400 Å^2^) (6). Rab21 uses switch I, switch II, and the interswitch region to interact with the substrate-binding site of Rabex-5. Although the binding sites for switch II and a small portion of the interswitch region of Rab21 are occupied by the three-helix bundle, the binding sites for switch I and a large portion of the interswitch region of Rab21 are exposed (Figure 1—figure supplement 3). Hence, we consider that the substrate-binding site of Rabex-5 is largely exposed to the solvent and partially accessible by the substrate.

In the previous Figure 1—figure supplement 3, we colored switch I, switch II, and the interswitch region of Rab21 in orange, yellow, and green, respectively, and the other regions in magenta and labeled Rab21 in magenta as well. That might lead the reviewers to consider the magenta colored region as the whole of Rab21. To avoid confusion, we have regenerated it as Figure 1—figure supplement 3 in a slightly different orientation and with different colorings. In this new figure, switch II and the interswitch region of Rab21 remain to be colored in dark yellow and green, respectively, but switch I is colored in red and the other regions in gray. Meanwhile, we color Rabex-5CC and Rabaptin-5C21 in violet and pink, respectively. In this new figure, it is clearer that switch I and a large portion of the interswitch region are not occupied by the three-helix bundle.

*7) Abstract: “Rabaptin-5C212” should be defined on first usage. In particular, clarify what the “2” after “Rabaptin-5C21” signifies*.

We thank the reviewers for this suggestion. In the Abstract, we have specifically spelled out the dimeric Rabaptin-5C21 as follows: “We report here the crystal structures of Rabex-5 in complex with the dimeric Rabaptin-5C21 (Rabaptin-5C21_2_) and Rabex-5 in complex with Rabaptin-5C21_2_ and Rab5.”

*8) Several abbreviations (e.g., R2, R2delta, R3, etc) are missing from the list. The authors should decrease jargon wherever possible as there are many acronyms here. Thus, please define R2 complex? R3 complex? Constructs used in 3G? It is hard for the reader*.

We introduced the abbreviations of R2, R2Δ, R3, and R3Δto avoid lengthy description of the complexes. As suggested by the reviewers, we have added the abbreviations for R2, R2Δ, R3, and R3Δ in the abbreviation list and defined these abbreviations when they are first used in the text. We have also avoided to using these abbreviations if ambiguity might exist.

The constructs used in Figure 5 (original Figure 3) have been specified in the Materials and methods section (Immunocytochemistry) as follows: “GFP-tagged Rab5 (residues 1-215), Myc-tagged Rabaptin-5 (residues 552-862), and Flag-tagged Rabex-5 (residues 132-455), Flag-tagged Rabex-5Δ (residues 132-455Δ393-407), and Flag-tagged Rabex-5 GEF (residues 132-392) were cloned into the pcDNA3 vector (Invitrogen), respectively.”

*9) Finally, because the assignment and orientation of side chains shown in*
Figure 2
*are determined primarily by the molecular replacement models and only a little bit by the data, please add a qualifying note about the uncertainty of the side chain orientations due to the low resolution of the structure*.

As suggested by the reviewers, we have noted the uncertainty of the side-chain orientations in the Materials and methods section (Crystallization, data collection, and structure determination) as follows: “Due to the low resolution of the diffraction data, the structure models of Rab5, Rabex-5, and Rabaptin-5C21 in the R3Δ complex were refined as rigid bodies with deformable elastic network and group B-factor restraints (Schroder et al, 2010) and thus the side-chain orientations in this complex are somewhat uncertain.”